# CAt-Walk: Inductive Hypergraph Learning via Set Walks

**Ali Behrouz**
Department of Computer Science
University of British Columbia
alibez@cs.ubc.ca

**Farnoosh Hashemi**[†]
Department of Computer Science
University of British Columbia
farsh@cs.ubc.ca

**Sadaf Sadeghian**[†]
Department of Computer Science
University of British Columbia
sadafsdn@cs.ubc.ca

**Margo Seltzer**
Department of Computer Science
University of British Columbia
mseltzer@cs.ubc.ca

## Abstract

Temporal hypergraphs provide a powerful paradigm for modeling time-dependent, higher-order interactions in complex systems. Representation learning for hypergraphs is essential for extracting patterns of the higher-order interactions that are critically important in real-world problems in social network analysis, neuroscience, finance, etc. However, existing methods are typically designed only for specific tasks or static hypergraphs. We present CAt-Walk, an inductive method that learns the underlying dynamic laws that govern the temporal and structural processes underlying a temporal hypergraph. CAt-Walk introduces a temporal, higher-order walk on hypergraphs, SetWalk, that extracts higher-order causal patterns. CAt-Walk uses a novel adaptive and permutation invariant pooling strategy, SetMixer, along with a set-based anonymization process that hides the identity of hyperedges. Finally, we present a simple yet effective neural network model to encode hyperedges. Our evaluation on 10 hypergraph benchmark datasets shows that CAt-Walk attains outstanding performance on temporal hyperedge prediction benchmarks in both inductive and transductive settings. It also shows competitive performance with state-of-the-art methods for node classification. (Code)

## 1 Introduction

Temporal networks have become increasingly popular for modeling interactions among entities in dynamic systems [1–5]. While most existing work focuses on pairwise interactions between entities, many real-world complex systems exhibit natural relationships among multiple entities [6–8]. Hypergraphs provide a natural extension to graphs by allowing an edge to connect any number of vertices, making them capable of representing higher-order structures in data. Representation learning on (temporal) hypergraphs has been recognized as an important machine learning problem and has become the cornerstone behind a wealth of high-impact applications in computer vision [9, 10], biology [11, 12], social networks [13, 14], and neuroscience [15, 16].

Many recent attempts to design representation learning methods for hypergraphs are equivalent to applying Graph Neural Networks (GNNs) to the clique-expansion (CE) of a hypergraph [17–21]. CE is a straightforward way to generalize graph algorithms to hypergraphs by replacing hyperedges with (weighted) cliques [18–20]. However, this decomposition of hyperedges limits expressiveness,

---

[†]These two authors contributed equally (ordered alphabetically) and reserve the right to swap their order.

37th Conference on Neural Information Processing Systems (NeurIPS 2023).

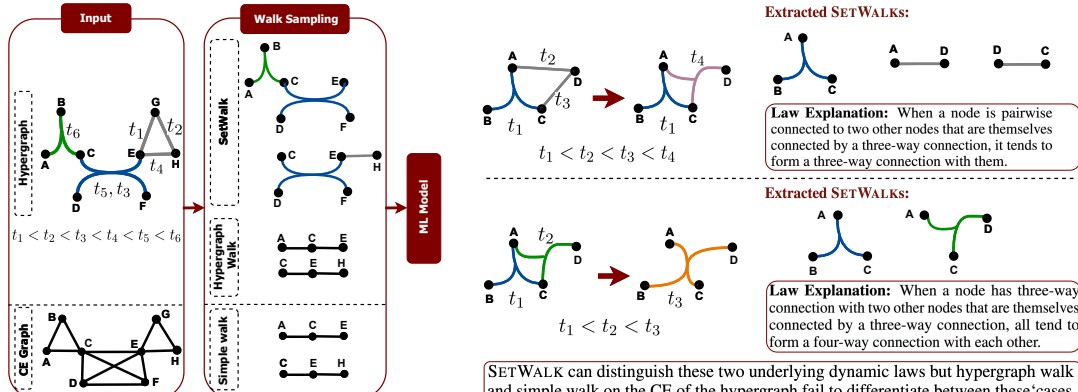

Figure 1: The advantage of SETWALKS in walk-based hypergraph learning.

Figure 2: The advantage of SETWALKS in causality extraction and capturing complex dynamic laws.

leading to suboptimal performance [6, 22–24] (see Theorem 1 and Theorem 4). New methods that encode hypergraphs directly partially address this issue [11, 25–28]. However, these methods suffer from some combination of the following three limitations: they are designed for ① learning the structural properties of *static hypergraphs* and do not consider temporal properties, ② the transductive setting, limiting their performance on unseen patterns and data, and ③ a specific downstream task (e.g., node classification [25], hyperedge prediction [26], or subgraph classification [27]) and cannot easily be extended to other downstream tasks, limiting their application.

Temporal motif-aware and neighborhood-aware methods have been developed to capture complex patterns in data [29–31]. However, counting temporal motifs in large networks is time-consuming and non-parallelizable, limiting the scalability of these methods. To this end, several recent studies suggest using temporal random walks to automatically retrieve such motifs [32–36]. One possible solution to capturing underlying temporal and higher-order structure is to extend the concept of a hypergraph random walk [37–43] to its temporal counterpart by letting the walker walk over time. However, existing definitions of random walks on hypergraphs offer limited expressivity and sometimes degenerate to simple walks on the CE of the hypergraph [40] (see Appendix C). There are two reasons for this: ① Random walks are composed of a sequence of *pair-wise* interconnected vertices, even though edges in a hypergraph connect *sets* of vertices. Decomposing them into sequences of simple pair-wise interactions loses the semantic meaning of the hyperedges (see Theorem 4). ② A sampling probability of a walk on a hypergraph must be different from its sampling probability on the CE of the hypergraph [37–43]. However, Chitra and Raphael [40] shows that each definition of the random walk with edge-independent sampling probability of nodes is equivalent to random walks on a weighted CE of the hypergraph. Existing studies on random walks on hypergraphs ignore ① and focus on ② to distinguish the walks on simple graphs and hypergraphs. However, as we show in Table 2, ① is equally important, if not more so.

For example, Figure 1 shows the procedure of existing walk-based machine learning methods on a temporal hypergraph. The neural networks in the model take as input only sampled walks. However, the output of the hypergraph walk [37, 38] and simple walk on the CE graph are the same. This means that the neural network cannot distinguish between pair-wise and higher-order interactions.

We present **C**ausal **A**nonymous Se**t** **Walk**s (CAT-WALK), an inductive hyperedge learning method. We introduce a hyperedge-centric random walk on hypergraphs, called SETWALK, that automatically extracts temporal, higher-order motifs. The hyperedge-centric approach enables SETWALKS to distinguish multi-way connections from their corresponding CEs (see Figure 1, Figure 2, and Theorem 1). We use temporal hypergraph motifs that reflect network dynamics (Figure 2) to enable CAT-WALK to work well in the inductive setting. To make the model agnostic to the hyperedge identities of these motifs, we use two-step, set-based anonymization: ① Hide node identities by assigning them new positional encodings based on the number of times that they appear at a specific position in a set of sampled SETWALKS, and ② Hide hyperedge identities by combining the positional encodings of the nodes comprising the hyperedge using a novel permutation invariant pooling strategy, called SETMIXER. We incorporate a neural encoding method that samples a few SETWALKS starting from nodes of interest. It encodes and aggregates them via MLP-MIXER [44] and our new pooling strategy

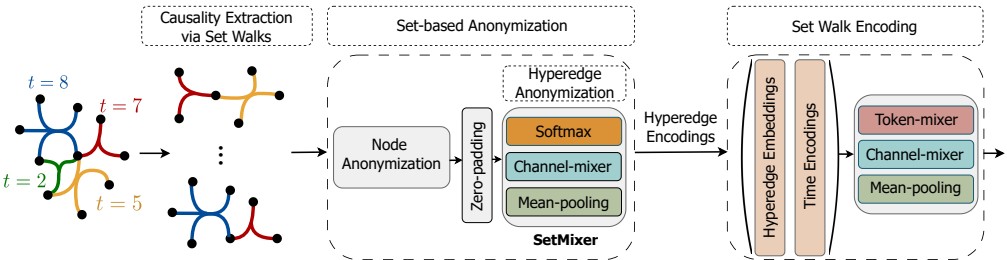

Figure 3: **Schematic of the CAT-WALK**. CAT-WALK consists of three stages: (1) Causality Extraction via Set Walks (§3.2), (2) Set-based Anonymization (§3.3), and (3) Set Walk Encoding (§3.4).

SETMIXER, respectively, to predict temporal, higher-order interactions. Finally, we discuss how to extend CAT-WALK for node classification. Figure 3 shows the schematic of the CAT-WALK.

We theoretically and experimentally discuss the effectiveness of CAT-WALK and each of its components. More specifically, we prove that SETWALKS are more expressive than existing random walk algorithms on hypergraphs. We demonstrate SETMIXER's efficacy as a permutation invariant pooling strategy for hypergraphs and prove that using it in our anonymization process makes that process more expressive than existing anonymization processes [33, 45, 46] when applied to the CE of the hypergraphs. To the best of our knowledge, we report the most extensive experiments in the hypergraph learning literature pertaining to unsupervised hyperedge prediction with 10 datasets and eight baselines. Results show that our method produces 9% and 17% average improvement in transductive and inductive settings, outperforming all state-of-the-art baselines in the hyperedge prediction task. Also, CAT-WALK achieves the best or on-par performance on dynamic node classification tasks. All proofs appear in the Appendix.

## 2 Related Work

Temporal graph learning is an active research area [5, 47]. A major group of methods uses GNNs to learn node encodings and Recurrent Neural Networks (RNNs) to update these encodings over time [48–55]. More sophisticated methods based on anonymous temporal random walks [33, 34], line graphs [56], GraphMixer [57], neighborhood representation [58], and subgraph sketching [59] are designed to capture complex structures in vertex neighborhoods. Although these methods show promising results in a variety of tasks, they are fundamentally limited in that they are designed for *pair-wise* interaction among vertices and not the higher-order interactions in hypergraphs.

Representation learning on hypergraphs addresses this problem [17, 60]. We group work in this area into three overlapping categories:

① **Clique and Star Expansion**: CE-based methods replace hyperedges with (weighted) cliques and apply GNNs, sometimes with sophisticated propagation rules [21, 25], degree normalization, and nonlinear hyperedge weights [17–21, 39, 61, 62]. Although these methods are simple, it is well-known that CE causes undesired losses in learning performance, specifically when relationships within an incomplete subset of a hyperedge do not exist [6, 22–24]. Star expansion (SE) methods first use hypergraph star expansion and model the hypergraph as a bipartite graph, where one set of vertices represents nodes and the other represents hyperedges [25, 28, 63, 64]. Next, they apply modified heterogeneous GNNs, possibly with dual attention mechanisms from nodes to hyperedges and vice versa [25, 27]. Although this group does not cause as large a distortion as CE, they are neither memory nor computationally efficient. ② **Message Passing**: Most existing hypergraph learning methods, use message passing over hypergraphs [17–21, 25–27, 39, 61, 62, 65, 66]. Recently, Chien et al. [25] and Huang and Yang [28] designed universal message-passing frameworks that include propagation rules of most previous methods (e.g., [17, 19]). The main drawback of these two frameworks is that they are limited to node classification tasks and do not easily generalize to other tasks. ③ **Walk-based**: random walks are a common approach to extracting graph information for machine learning algorithms [32–34, 67, 68]. Several walk-based hypergraph learning methods are designed for a wide array of applications [43, 69–78]. However, most existing methods use simple random walks on the CE of the hypergraph (e.g., [26, 43, 78]). More complicated random walks on hypergraphs address this limitation [40–42, 79]. Although some of these studies show that their walk's transition matrix

differs from the simple walk on the CE [40, 79], their extracted walks can still be the same, limiting their expressivity (see Figure 1, Figure 2, and Theorem 1).

Our method differs from all prior (temporal) (hyper)graph learning methods in five ways: CAt-Walk: ① Captures temporal higher-order properties in a streaming manner: In contrast to existing methods in hyperedge prediction, our method captures temporal properties in a streaming manner, avoiding the drawbacks of snapshot-based methods. ② Works in the inductive setting by extracting underlying dynamic laws of the hypergraph, making it generalizable to unseen patterns and nodes. ③ Introduces a hyperedge-centric, temporal, higher-order walk with a new perspective on the walk sampling procedure. ④ Presents a new two-step anonymization process: Anonymization of higher-order patterns (i.e., SetWalks), requires hiding the identity of both nodes and hyperedges. We present a new permutation-invariant pooling strategy to hide hyperedges' identity according to their vertices, making the process provably more expressive than the existing anonymization processes [33, 34, 78]. ⑤ Removes self-attention and Rnns from the walk encoding procedure, avoiding their limitations. Appendices A and B provide a more comprehensive discussion of related work.

## 3 Method: CAt-Walk Network

### 3.1 Preliminaries

**Definition 1** (Temporal Hypergraphs). *A temporal hypergraph $\mathcal{G} = (\mathcal{V}, \mathcal{E}, \mathcal{X})$, can be represented as a sequence of hyperedges that arrive over time, i.e., $\mathcal{E} = \{(e_1, t_1), (e_2, t_2), \dots\}$, where $\mathcal{V}$ is the set of nodes, $e_i \in 2^{\mathcal{V}}$ are hyperedges, $t_i$ is the timestamp showing when $e_i$ arrives, and $\mathcal{X} \in \mathbb{R}^{|\mathcal{V}| \times f}$ is a matrix that encodes node attribute information for nodes in $\mathcal{V}$. Note that we treat each hyperedge $e_i$ as the set of all vertices connected by $e_i$.*

**Example 1.** *The input of Figure 1 shows an example of a temporal hypergraph. Each hyperedge is a set of nodes that are connected by a higher-order connection. Each higher-order connection is specified by a color (e.g., green, blue, and gray), and also is associated with a timestamp (e.g., $t_i$).*

Given a hypergraph $\mathcal{G} = (\mathcal{V}, \mathcal{E}, \mathcal{X})$, we represent the set of hyperedges attached to a node $u$ before time $t$ as $\mathcal{E}^t(u) = \{(e, t') | t' < t, u \in e\}$. We say two hyperedges $e$ and $e'$ are adjacent if $e \cap e' \neq \emptyset$ and use $\mathcal{E}^t(e) = \{(e', t') | t' < t, e' \cap e \neq \emptyset\}$ to represent the set of hyperedges adjacent to $e$ before time $t$. Next, we focus on the hyperedge prediction task: Given a subset of vertices $\mathcal{U} = \{u_1, \dots, u_k\}$, we want to predict whether a hyperedge among all the $u_i$s will appear in the next timestamp or not. In AppendixG.2 we discuss how this approach can be extended to node classification.

The (hyper)graph isomorphism problem is a decision problem that decides whether a pair of (hyper)graphs are isomorphic or not. Based on this concept, next, we discuss the measure we use to compare the expressive power of different methods.

**Definition 2** (Expressive Power Measure). *Given two methods $\mathcal{M}_1$ and $\mathcal{M}_2$, we say method $\mathcal{M}_1$ is more expressive than method $\mathcal{M}_2$ if:*

> *1. for any pair of (hyper)graphs $(\mathcal{G}_1, \mathcal{G}_2)$ such that $\mathcal{G}_1 \neq \mathcal{G}_2$, if method $\mathcal{M}_1$ can distinguish $\mathcal{G}_1$ and $\mathcal{G}_2$ then method $\mathcal{M}_2$ can also distinguish them,*
> *2. there is a pair of hypergraphs $\mathcal{G}'_1 \neq \mathcal{G}'_2$ such that $\mathcal{M}_1$ can distinguish them but $\mathcal{M}_2$ cannot.*

### 3.2 Causality Extraction via SetWalk

The collaboration network in Figure 1 shows how prior work that models hypergraph walks as sequences of vertices fails to capture complex connections in hypergraphs. Consider the two walks: $A \to C \to E$ and $H \to E \to C$. These two walks can be obtained either from a hypergraph random walk or from a simple random walk on the CE graph. Due to the symmetry of these walks with respect to $(A, C, E)$ and $(H, E, C)$, they cannot distinguish $A$ and $H$, although the neighborhoods of these two nodes exhibit different patterns: $A, B$, and $C$ have published a paper together (connected by a hyperedge), but each pair of $E, G$, and $H$ has published a paper (connected by a pairwise link). We address this limitation by defining a temporal walk on hypergraphs as a sequence of hyperedges:

**Definition 3** (SetWalk). *Given a temporal hypergraph $\mathcal{G} = (\mathcal{V}, \mathcal{E}, \mathcal{X})$, a SetWalk with length $\ell$ on temporal hypergraph $\mathcal{G}$ is a randomly generated sequence of hyperedges (sets):*

$$\text{Sw} : (e_1, t_{e_1}) \to (e_2, t_{e_2}) \to \cdots \to (e_\ell, t_{e_\ell}),$$

where $e_i \in \mathcal{E}$, $t_{e_{i+1}} < t_{e_i}$, and the intersection of $e_i$ and $e_{i+1}$ is not empty, $e_i \cap e_{i+1} \neq \emptyset$. In other words, for each $1 \leq i \leq \ell - 1$: $e_{i+1} \in \mathcal{E}^{t_i}(e_i)$. We use Sw[i] to denote the i-th hyperedge-time pair in the SETWALK. That is, Sw[i][0] = $e_i$ and Sw[i][1] = $t_{e_i}$.

**Example 2.** *Figure 1 illustrates a temporal hypergraph with two sampled SETWALKs, hypergraph random walks, and simple walks. As an example, $(\{A, B, C\}, t_6) \rightarrow (\{C, D, E, F\}, t_5)$ is a SETWALK that starts from hyperedge e including $\{A, B, C\}$ in time $t_6$, backtracks overtime and then samples hyperedge $e'$ including $\{C, D, E, F\}$ in time $t_5 < t_6$. While this higher-order random walk with length two provides information about all $\{A, B, C, D, E, F\}$, its simple hypergraph walk counterpart, i.e. $A \rightarrow C \rightarrow E$, provides information about only three nodes.*

Next, we formally discuss the power of SETWALKs. The proof of the theorem can be found in Appendix E.1.

**Theorem 1.** *A random SETWALK is equivalent to neither the hypergraph random walk, the random walk on the CE graph, nor the random walk on the SE graph. Also, for a finite number of samples of each, SETWALK is more expressive than existing walks.*

In Figure 1, SETWALKs capture higher-order interactions and distinguish the two nodes $A$ and $H$, which are indistinguishable via hypergraph random walks and graph random walks in the CE graph. We present a more detailed discussion and comparison with previous definitions of random walks on hypergraphs in Appendix C.

**Causality Extraction.** We introduce a sampling method to allow SETWALKs to extract temporal higher-order motifs that capture causal relationships by backtracking over time and sampling adjacent hyperedges. As discussed in previous studies [33, 34], more recent connections are usually more informative than older connections. Inspired by Wang et al. [33], we use a hyperparameter $\alpha \geq 0$ to sample a hyperedge $e$ with probability proportional to $\exp(\alpha(t - t_p))$, where $t$ and $t_p$ are the timestamps of $e$ and the previously sampled hyperedge in the SETWALK, respectively. Additionally, we want to bias sampling towards pairs of adjacent hyperedges that have a greater number of common nodes to capture higher-order motifs. However, as discussed in previous studies, the importance of each node for each hyperedge can be different [25, 27, 40, 65]. Accordingly, the transferring probability from hyperedge $e_i$ to its adjacent hyperedge $e_j$ depends on the importance of the nodes that they share. We address this via a temporal SETWALK sampling process with *hyperedge-dependent node weights*. Given a temporal hypergraph $\mathcal{G} = (\mathcal{V}, \mathcal{E}, \mathcal{X})$, a hyperedge-dependent node-weight function $\Gamma : \mathcal{V} \times \mathcal{E} \rightarrow \mathbb{R}^{\geq 0}$, and a previously sampled hyperedge $(e_p, t_p)$, we sample a hyperedge $(e, t)$ with probability:

$$\mathbb{P}[(e, t)|(e_p, t_p)] \propto \frac{\exp(\alpha(t - t_p))}{\sum_{(e', t') \in \mathcal{E}^{t_p}(e_p)} \exp(\alpha(t' - t_p))} \times \frac{\exp(\varphi(e, e_p))}{\sum_{(e', t') \in \mathcal{E}^{t_p}(e_p)} \exp(\varphi(e', e_p))}, \tag{1}$$

where $\varphi(e, e') = \sum_{u \in e \cap e'} \Gamma(u, e)\Gamma(u, e')$, representing the assigned weight to $e \cap e'$. We refer to the first and second terms as *temporal bias* and *structural bias*, respectively.

The pseudocode of our SETWALK sampling algorithm and its complexity analysis are in Appendix D. We also discuss this hyperedge-dependent sampling procedure and how it is provably more expressive than existing hypergraph random walks in Appendix C.

Given a (potential) hyperedge $e_0 = \{u_1, u_2, \ldots, u_k\}$ and a time $t_0$, we say a SETWALK, Sw, starts from $u_i$ if $u_i \in$ Sw[1][0]. We use the above procedure to generate $M$ SETWALKs with length $m$ starting from each $u_i \in e_0$. We use $\mathcal{S}(u_i)$ to store SETWALKs that start from $u_i$.

### 3.3 Set-based Anonymization of Hyperedges

In the anonymization process, we replace hyperedge identities with position encodings, capturing structural information while maintaining the inductiveness of the method. Micali and Zhu [45] studied Anonymous Walks (AWs), which replace a *node's identity* by the order of its appearance in each walk. The main limitation of AWs is that the position encoding of each node depends only on its specific walk, missing the dependency and correlation of different sampled walks [33]. To mitigate this drawback, Wang et al. [33] suggest replacing node identities with the hitting counts of the nodes based on a set of sampled walks. In addition to the fact that this method is designed for walks on simple graphs, there are two main challenges to adopting it for SETWALKs: ① SETWALKs

are a sequence of hyperedges, so we need an encoding for the position of hyperedges. Natural attempts to replace hyperedges' identity with the hitting counts of the hyperedges based on a set of sampled SETWALKS, misses the similarity of hyperedges with many of the same nodes. ② Each hyperedge is a set of vertices and natural attempts to encode its nodes' positions and aggregate them to obtain a position encoding of the hyperedge requires a permutation invariant pooling strategy. This pooling strategy also requires consideration of the higher-order dependencies between obtained nodes' position encodings to take advantage of higher-order interactions (see Theorem 2). To address these challenges we present a set-based anonymization process for SETWALKS. Given a hyperedge $e_0 = \{u_1, \ldots, u_k\}$, let $w_0$ be any node in $e_0$. For each node $w$ that appears on at least one SETWALK in $\bigcup_{i=1}^{k} \mathcal{S}(u_i)$, we assign a relative, node-dependent node identity, $\mathcal{R}(w, \mathcal{S}(w_0)) \in \mathbb{Z}^m$, as follows:

$$\mathcal{R}(w, \mathcal{S}(w_0))[i] = |\{\text{Sw}|\text{Sw} \in \mathcal{S}(w_0), w \in \text{Sw}[i][0]\}| \quad \forall i \in \{1, 2, \ldots, m\}. \tag{2}$$

For each node $w$ we further define $\text{ID}(w, e_0) = \{\mathcal{R}(w, \mathcal{S}(u_i))\}_{i=1}^{k}$. Let $\Psi(.) : \mathbb{R}^{d \times d_1} \to \mathbb{R}^{d_2}$ be a pooling function that gets a set of $d_1$-dimensional vectors and aggregates them to a $d_2$-dimensional vector. Given two instances of this pooling function, $\Psi_1(.)$ and $\Psi_2(.)$, for each hyperedge $e = \{w_1, w_2, \ldots, w_{k'}\}$ that appears on at least one SETWALK in $\bigcup_{i=1}^{k} \mathcal{S}(u_i)$, we assign a relative hyperedge identity as:

$$\text{ID}(e, e_0) = \Psi_1 \left( \{\Psi_2 (\text{ID}(w_i, e_0))\}_{i=1}^{k'} \right). \tag{3}$$

That is, for each node $w_i \in e$ we first aggregate its relative node-dependent identities (i.e., $\mathcal{R}(w_i, \mathcal{S}(u_j))$) to obtain the relative hyperedge-dependent identity. Then we aggregate these hyperedge-dependent identities for all nodes in $e$. Since the size of hyperedges can vary, we zero-pad to a fixed length. Note that this zero padding is important to capture the size of the hyperedge. The hyperedge with more zero-padded dimensions has fewer nodes.

This process addresses the first challenge and encodes the position of hyperedges. Unfortunately, many simple and known pooling strategies (e.g., SUM(.), ATTN-SUM(.), MEAN(.), etc.) can cause missing information when applied to hypergraphs. We formalize this in the following theorem:

**Theorem 2.** *Given an arbitrary positive integer $k \in \mathbb{Z}^+$, let $\Psi(.)$ be a pooling function such that for any set $S = \{w_1, \ldots, w_d\}$:*

$$\Psi(S) = \sum_{\substack{S' \subseteq S \\ |S'| = k}} f(S'), \tag{4}$$

*where $f$ is some function. Then the pooling function can cause missing information, meaning that it limits the expressiveness of the method to applying to the projected graph of the hypergraph.*

While simple concatenation does not suffer from this undesirable property, it is not permutation invariant. To overcome these challenges, we design an all-MLP permutation invariant pooling function, SETMIXER, that not only captures higher-order dependencies of set elements but also captures dependencies across the number of times that a node appears at a certain position in SETWALKS.

**SETMIXER.** MLP-MIXER [44] is a family of models based on multi-layer perceptrons, widely used in the computer vision community, that are simple, amenable to efficient implementation, and robust to over-squashing and long-term dependencies (unlike RNNS and attention mechanisms) [44, 57]. However, the token-mixer phase of these methods is sensitive to the order of the input (see Appendix A). To address this limitation, inspired by MLP-MIXER [44], we design SETMIXER as follows: Let $S = \{\mathbf{v}_1, \ldots, \mathbf{v}_d\}$, where $\mathbf{v}_i \in \mathbb{R}^{d_1}$, be the input set and $\mathbf{V} = [\mathbf{v}_1, \ldots, \mathbf{v}_d] \in \mathbb{R}^{d \times d_1}$ be its matrix representation:

$$\Psi(S) = \text{MEAN} \left( \mathbf{H}_{\text{token}} + \sigma \left( \text{LayerNorm} (\mathbf{H}_{\text{token}}) \mathbf{W}_s^{(1)} \right) \mathbf{W}_s^{(2)} \right), \tag{5}$$

where

$$\mathbf{H}_{\text{token}} = \mathbf{V} + \sigma \left( \text{Softmax} \left( \text{LayerNorm} (\mathbf{V})^T \right) \right)^T. \tag{6}$$

Here, $\mathbf{W}_s^{(1)}$ and $\mathbf{W}_s^{(2)}$ are learnable parameters, $\sigma(.)$ is an activation function (we use GeLU [80] in our experiments), and LayerNorm is layer normalization [81]. Equation 5 is the channel mixer and Equation 6 is the token mixer. The main intuition of SETMIXER is to use the Softmax(.) function to bind token-wise information in a non-parametric manner, avoiding permutation variant operations in the token mixer. We formally prove the following theorem in Appendix E.3.

**Theorem 3.** SETMIXER *is permutation invariant and is a universal approximator of invariant multi-set functions. That is,* SETMIXER *can approximate any invariant multi-set function.*

Based on the above theorem, SETMIXER can overcome the challenges we discussed earlier as it is permutation invariant. Also, it is a universal approximator of multi-set functions, which shows its power to learn any arbitrary function. Accordingly, in our anonymization process, we use $\Psi(.) = $ SETMIXER$(.)$ in Equation 3 to hide hyperedge identities. Next, we guarantee that our anonymization process does not depend on hyperedges or nodes identities, which justifies the claim of inductiveness of our model:

**Proposition 1.** *Given two (potential) hyperedges $e_0 = \{u_1, \ldots, u_k\}$ and $e'_0 = \{u'_1, \ldots, u'_k\}$, if there exists a bijective mapping $\pi$ between node identities such that for each SETWALK like $Sw \in \bigcup_{i=1}^{k} \mathcal{S}(u_i)$ can be mapped to one SETWALK like $Sw' \in \bigcup_{i=1}^{k} \mathcal{S}(u'_i)$, then for each hyperedge $e = \{w_1, \ldots, w_{k'}\}$ that appears in at least one SETWALK in $\bigcup_{i=1}^{k} \mathcal{S}(u_i)$, we have $\text{ID}(e, e_0) = \text{ID}(\pi(e), e'_0)$, where $\pi(e) = \{\pi(w_1), ,\ldots, \pi(w_{k'})\}$.*

Finally, we guarantee that our anonymization approach is more expressive than existing anonymization process [33, 45] when applied to the CE of the hypergraphs:

**Theorem 4.** *The set-based anonymization method is more expressive than any existing anonymization strategies on the CE of the hypergraph. More precisely, there exists a pair of hypergraphs $\mathcal{G}_1 = (\mathcal{V}_1, \mathcal{E}_1)$ and $\mathcal{G}_2 = (\mathcal{V}_2, \mathcal{E}_2)$ with different structures (i.e., $\mathcal{G}_1 \not\cong \mathcal{G}_2$) that are distinguishable by our anonymization process and are not distinguishable by the CE-based methods.*

### 3.4  SETWALK Encoding

Previous walk-based methods [33, 34, 78] view a walk as a sequence of nodes. Accordingly, they plug nodes' positional encodings in a RNN [82] or Transformer [83] to obtain the encoding of each walk. However, in addition to the computational cost of RNN and Transformers, they suffer from over-squashing and fail to capture long-term dependencies. To this end, we design a simple and low-cost SETWALK encoding procedure that uses two steps: ① A time encoding module to distinguish different timestamps, and ② A mixer module to summarize temporal and structural information extracted by SETWALKS.

**Time Encoding.** We follow previous studies [33, 84] and adopt random Fourier features [85, 86] to encode time. However, these features are periodic, so they capture only periodicity in the data. We add a learnable linear term to the feature representation of the time encoding. We encode a given time $t$ as follows:

$$\text{T}(t) = (\boldsymbol{\omega}_l t + \mathbf{b}_l) \| \cos(t\boldsymbol{\omega}_p), \tag{7}$$

where $\boldsymbol{\omega}_l, \mathbf{b}_l \in \mathbb{R}$ and $\boldsymbol{\omega}_p \in \mathbb{R}^{d_2 - 1}$ are learnable parameters and $\|$ shows concatenation.

**MIXER Module.** To summarize the information in each SETWALK, we use a MLP-MIXER [44] on the sequence of hyperedges in a SETWALK as well as their corresponding encoded timestamps. Contrary to the anonymization process, where we need a permutation invariant procedure, here, we need a permutation variant procedure since the order of hyperedges in a SETWALK is important. Given a (potential) hyperedge $e_0 = \{u_1, \ldots, u_k\}$, we first assign $\text{ID}(e, e_0)$ to each hyperedge $e$ that appears on at least one sampled SETWALK starting from $e_0$ (Equation 3). Given a SETWALK, $Sw : (e_1, t_{e_1}) \rightarrow \cdots \rightarrow (e_m, t_{e_m})$, we let $\mathbf{E}$ be a matrix that $\mathbf{E}_i = \text{ID}(e_i, e_0) \| \text{T}(t_{e_i})$:

$$\text{ENC}(Sw) = \text{MEAN}\left(\mathbf{H}_{\text{token}} + \sigma\left(\texttt{LayerNorm}\left(\mathbf{H}_{\text{token}}\right)\mathbf{W}_{\text{channel}}^{(1)}\right)\mathbf{W}_{\text{channel}}^{(2)}\right), \tag{8}$$

where

$$\mathbf{H}_{\text{token}} = \mathbf{E} + \mathbf{W}_{\text{token}}^{(2)}\sigma\left(\mathbf{W}_{\text{token}}^{(1)}\texttt{LayerNorm}\left(\mathbf{E}\right)\right). \tag{9}$$

### 3.5  Training

In the training phase, for each hyperedge in the training set, we adopt the commonly used negative sample generation method [60] to generate a negative sample. Next, for each hyperedge in the training set such as $e_0 = \{u_1, u_2, \ldots, u_k\}$, including both positive and negative samples, we sample $M$ SETWALKs with length $m$ starting from each $u_i \in e_0$ to construct $\mathcal{S}(u_i)$. Next, we anonymize each hyperedge that appears in at least one SETWALK in $\bigcup_{i=1}^{k} \mathcal{S}(u_i)$ by Equation 3 and then use the MIXER module to encode each $Sw \in \bigcup_{i=1}^{k} \mathcal{S}(u_i)$. To encode each node $u_i \in e_0$, we use MEAN$(.)$ pooling over SETWALKs in $\mathcal{S}(u_i)$. Finally, to encode $e_0$ we use SETMIXER to mix obtained node encodings. For

hyperedge prediction, we use a 2-layer perceptron over the hyperedge encodings to make the final prediction. We discuss node classification in Appendix G.2.

## 4 Experiments

We evaluate the performance of our model on two important tasks: hyperedge prediction and node classification (see Appendix G.2) in both inductive and transductive settings. We then discuss the importance of our model design and the significance of each component in CAt-Walk.

### 4.1 Experimental Setup

**Baselines.** We compare our method to eight previous state-of-the-art baselines on the hyperedge prediction task. These methods can be grouped into three categories: ① Deep hypergraph learning methods including HyperSAGCN [26], NHP [87], and CHESHIRE [11]. ② Shallow methods including HPRA [88] and HPLSF [89]. ③ CE methods: CE-CAW [33], CE-EvolveGCN [90] and CE-GCN [91]. Details on these models and hyperparameters used appear in Appendix F.2.

**Datasets.** We use 10 available benchmark datasets [6] from the existing hypergraph neural networks literature. These datasets' domains include drug networks (i.e., NDC [6]), contact networks (i.e., High School [92] and Primary School [93]), the US. Congress bills network [94, 95], email networks (i.e., Email Enron [6] and Email Eu [96]), and online social networks (i.e., Question Tags and Users-Threads [6]). Detailed descriptions of these datasets appear in Appendix F.1.

**Evaluation Tasks.** We focus on Hyperedge Prediction: In the transductive setting, we train on the temporal hyperedges with timestamps less than or equal to $T_{\text{train}}$ and test on those with timestamps greater than $T_{\text{train}}$. Inspired by Wang et al. [33], we consider two inductive settings. In the **Strongly Inductive** setting, we predict hyperedges consisting of some unseen nodes. In the **Weakly Inductive** setting, we predict hyperedges with *at least* one seen and some unseen nodes. We first follow the procedure used in the transductive setting, and then we randomly select 10% of the nodes and remove all hyperedges that include them from the training set. We then remove all hyperedges with seen nodes from the validation and testing sets. For dynamic node classification, see Appendix G.2. For all datasets, we fix $T_{\text{train}} = 0.7\,T$, where $T$ is the last timestamp. To evaluate the models' performance we follow the literature and use Area Under the ROC curve (AUC) and Average Precision (AP).

### 4.2 Results and Discussion

**Hyperedge Prediction.** We report the results of CAt-Walk and baselines in Table 1 and Appendix G. The results show that CAt-Walk achieves the best overall performance compared to the baselines in both transductive and inductive settings. In the transductive setting, not only does our method outperform baselines in all but one dataset, but it achieves near perfect results (i.e., $\approx 98.0$) on the NDC and Primary School datasets. In the Weakly Inductive setting, our model achieves high scores (i.e., $> 91.5$) in all but one dataset, while most baselines perform poorly as they are not designed for the inductive setting and do not generalize well to unseen nodes or patterns. In the Strongly Inductive setting, CAt-Walk still achieves high AUC (i.e., $> 90.0$) on most datasets and outperforms baselines on *all* datasets. There are three main reasons for CAt-Walk's superior performance: ① Our SetWalks capture higher-order patterns. ② CAt-Walk incorporates temporal properties (both from SetWalks and our time encoding module), thus learning underlying dynamic laws of the network. The other temporal methods (CE-CAW and CE-EvolveGCN) are CE-based methods, limiting their ability to capture higher-order patterns. ③ CAt-Walk 's set-based anonymization process that avoids using node and hyperedge identities allows it to generalize to unseen patterns and nodes.

**Ablation Studies.** We next conduct ablation studies on the High School, Primary School, and Users-Threads datasets to validate the effectiveness of CAt-Walk's critical components. Table 2 shows AUC results for inductive hyperedge prediction. The first row reports the performance of the complete CAt-Walk implementation. Each subsequent row shows results for CAt-Walk with one module modification: row 2 replace SetWalk by edge-dependent hypergraph walk [40], row 3 removes the time encoding module, row 4 replaces SetMixer with Mean(.) pooling, row 5 replaces the SetMixer with sum-based universal approximator for sets [97], row 6 replaces the MLP-Mixer

Table 1: Performance on hyperedge prediction: Mean AUC (%) ± standard deviation. Boldfaced letters shaded blue indicate the best result, while gray shaded boxes indicate results within one standard deviation of the best result. N/A: the method has numerical precision or computational issues.

| | Methods | NDC Class | High School | Primary School | Congress Bill | Email Enron | Email Eu | Question Tags | Users-Threads |
|---|---|---|---|---|---|---|---|---|---|
| | | | | | **Strongly Inductive** | | | | |
| Inductive | CE-GCN | 52.31 ± 2.99 | 60.54 ± 2.06 | 52.34 ± 2.75 | 49.18 ± 3.61 | 63.04 ± 1.80 | 52.76 ± 2.41 | 56.10 ± 1.88 | 57.91 ± 1.56 |
| | CE-EvolveGCN | 49.78 ± 3.13 | 46.12 ± 3.83 | 58.01 ± 2.56 | 54.00 ± 1.84 | 57.31 ± 4.19 | 44.16 ± 1.27 | 64.08 ± 2.75 | 52.00 ± 2.32 |
| | CE-CAW | 76.45 ± 0.29 | 83.73 ± 1.42 | 80.31 ± 1.46 | 75.38 ± 1.25 | 70.81 ± 1.13 | 72.99 ± 0.20 | 70.14 ± 1.89 | 73.12 ± 1.06 |
| | NHP | 70.53 ± 4.95 | 65.29 ± 3.80 | 70.86 ± 3.42 | 69.82 ± 2.19 | 49.71 ± 6.09 | 65.35 ± 2.07 | 68.23 ± 3.34 | 71.83 ± 2.64 |
| | HyperSAGCN | 79.05 ± 2.48 | 88.12 ± 3.01 | 80.13 ± 1.38 | 79.51 ± 1.27 | 73.09 ± 2.60 | 78.01 ± 1.24 | 73.66 ± 1.95 | 73.94 ± 2.57 |
| | CHESHIRE | 72.24 ± 2.63 | 82.54 ± 0.88 | 77.26 ± 1.01 | 79.43 ± 1.58 | 70.03 ± 2.55 | 69.98 ± 2.71 | N/A | 76.99 ± 2.82 |
| | CAt-Walk | **98.89 ± 1.82** | **96.03 ± 1.50** | **95.32 ± 0.89** | **93.54 ± 0.56** | **73.45 ± 2.92** | **91.68 ± 2.78** | **88.03 ± 3.38** | **89.84 ± 6.02** |
| | | | | | **Weakly Inductive** | | | | |
| | CE-GCN | 51.80 ± 3.29 | 50.33 ± 3.40 | 52.19 ± 2.54 | 52.38 ± 2.75 | 50.81 ± 2.87 | 49.60 ± 3.96 | 55.13 ± 2.76 | 57.06 ± 3.16 |
| | CE-EvolveGCN | 55.39 ± 5.16 | 57.85 ± 3.51 | 51.50 ± 4.07 | 55.63 ± 3.41 | 45.66 ± 2.10 | 52.44 ± 2.38 | 61.79 ± 1.63 | 55.81 ± 2.54 |
| | CE-CAW | 77.61 ± 1.05 | 83.77 ± 1.41 | 82.98 ± 1.06 | 79.51 ± 0.94 | 80.54 ± 1.02 | 73.54 ± 1.19 | 77.29 ± 0.86 | 80.79 ± 0.82 |
| | NHP | 75.17 ± 2.02 | 67.25 ± 5.19 | 71.92 ± 1.83 | 69.58 ± 4.07 | 60.38 ± 4.45 | 67.19 ± 4.33 | 70.46 ± 3.52 | 76.44 ± 1.90 |
| | HyperSAGCN | 79.45 ± 2.18 | 88.53 ± 1.26 | 85.08 ± 1.45 | 80.12 ± 2.00 | 78.86 ± 0.63 | 77.26 ± 2.09 | 78.15 ± 1.41 | 75.38 ± 1.43 |
| | CHESHIRE | 79.03 ± 1.24 | 88.40 ± 1.06 | 83.55 ± 1.27 | 79.67 ± 0.83 | 74.53 ± 0.91 | 77.31 ± 0.95 | N/A | 81.27 ± 0.85 |
| | CAt-Walk | **99.16 ± 1.08** | **94.68 ± 2.37** | **96.53 ± 1.39** | **98.38 ± 0.21** | 64.11 ± 7.96 | **91.98 ± 2.41** | **90.28 ± 2.81** | **97.15 ± 1.81** |
| Transductive | HPRA | 70.83 ± 0.01 | 94.91 ± 0.00 | 89.86 ± 0.06 | 79.48 ± 0.03 | 78.62 ± 0.00 | 72.51 ± 0.00 | 83.18 ± 0.00 | 70.49 ± 0.02 |
| | HPLSF | 76.19 ± 0.82 | 92.14 ± 0.29 | 88.57 ± 1.09 | 79.31 ± 0.52 | 75.73 ± 0.05 | 75.27 ± 0.31 | 83.45 ± 0.93 | 74.38 ± 1.11 |
| | CE-GCN | 66.83 ± 3.74 | 62.99 ± 3.02 | 59.14 ± 3.87 | 64.42 ± 3.11 | 58.06 ± 3.80 | 64.19 ± 2.79 | 55.18 ± 5.12 | 62.78 ± 2.69 |
| | CE-EvolveGCN | 67.08 ± 3.51 | 65.19 ± 2.26 | 63.15 ± 1.32 | 69.30 ± 2.27 | 69.98 ± 5.38 | 64.36 ± 4.17 | 72.56 ± 1.72 | 68.55 ± 2.26 |
| | CE-CAW | 76.30 ± 0.84 | 81.63 ± 0.97 | 86.53 ± 0.84 | 76.99 ± 1.02 | 79.57 ± 0.14 | 78.19 ± 1.10 | 81.73 ± 2.48 | 80.86 ± 0.45 |
| | NHP | 82.39 ± 2.81 | 76.85 ± 3.08 | 80.04 ± 3.42 | 82.84 ± 1.61 | 80.27 ± 2.53 | 63.17 ± 3.79 | 79.14 ± 3.36 | 82.33 ± 1.02 |
| | HyperSAGCN | 80.76 ± 2.64 | 94.98 ± 1.30 | 90.77 ± 2.05 | 82.84 ± 1.61 | 83.59 ± 0.98 | 79.61 ± 2.35 | 84.07 ± 2.50 | 79.62 ± 2.04 |
| | CHESHIRE | 84.91 ± 1.05 | 95.11 ± 0.94 | 91.62 ± 1.18 | 86.81 ± 1.24 | 82.27 ± 0.86 | 86.38 ± 1.23 | N/A | 82.75 ± 1.99 |
| | CAt-Walk | **98.72 ± 1.38** | **95.30 ± 0.43** | **97.91 ± 3.30** | **88.15 ± 1.46** | 80.45 ± 5.30 | **96.74 ± 1.28** | **91.63 ± 1.41** | **93.51 ± 1.27** |

Table 2: Ablation study on CAt-Walk. AUC scores on inductive hyperedge prediction.

| | Methods | High School | Primary School | Users in Threads | Congress bill | Question Tags U |
|---|---|---|---|---|---|---|
| 1 | CAt-Walk | **96.03 ± 1.50** | **95.32 ± 0.89** | **89.84 ± 6.02** | **93.54 ± 0.56** | **97.59 ± 2.21** |
| 2 | Replace SetWalk by Random Walk | 92.10 ± 2.18 | 51.56 ± 5.63 | 53.24 ± 1.73 | 80.27 ± 0.02 | 67.74 ± 2.92 |
| 3 | Remove Time Encoding | 95.94 ± 0.19 | 86.80 ± 6.33 | 70.58 ± 9.32 | 92.56 ± 0.49 | 96.91 ± 1.89 |
| 4 | Replace SetMixer by Mean(.) | 94.58 ± 1.22 | 95.14 ± 4.36 | 63.59 ± 5.26 | 91.06 ± 0.24 | 68.62 ± 1.25 |
| 5 | Replace SetMixer by Sum-based | | | | | |
| 6 | Universal Approximator for Sets | 94.77 ± 0.67 | 90.86 ± 0.57 | 60.03 ± 1.16 | 91.07 ± 0.70 | 89.76 ± 0.45 |
| 7 | Replace MLP-Mixer by Rnn | 92.85 ± 1.53 | 50.29 ± 4.07 | 58.11 ± 1.60 | 54.90 ± 0.50 | 65.18 ± 1.99 |
| 8 | Replace MLP-Mixer by Transformer | 55.98 ± 0.83 | 86.64 ± 3.55 | 60.65 ± 1.56 | 89.38 ± 1.66 | 56.16 ± 4.03 |
| 9 | Fix $\alpha = 0$ | 74.06 ± 14.9 | 58.3 ± 18.62 | 74.41 ± 10.69 | 93.31 ± 0.13 | 62.41 ± 4.34 |

module with a Rnn (see  for more experiments on the significance of using MLP-Mixer in walk encoding), row 7 replaces the MLP-Mixer module with a Transformer [83], and row 8 replaces the hyperparameter $\alpha$ with uniform sampling of hyperedges over all time periods. These results show that each component is critical for achieving CAt-Walk's superior performance. The greatest contribution comes from SetWalk, MLP-Mixer in walk encoding, $\alpha$ in temporal hyperedge sampling, and SetMixer pooling, respectively.

**Hyperparameter Sensitivity.** We analyze the effect of hyperparameters used in CAt-Walk, including temporal bias coefficient $\alpha$, SetWalk length $m$, and sampling number $M$. The mean AUC performance on all inductive test hyperedges is reported in Figure 4. As expected, the left figure shows that

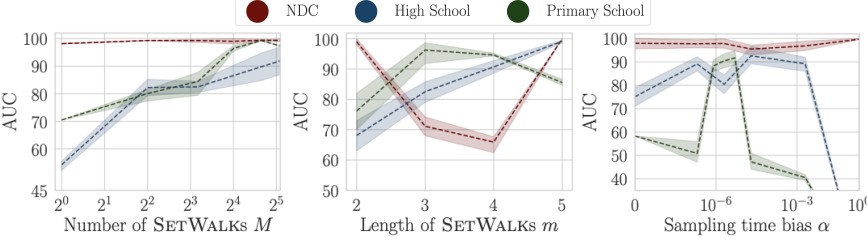

Figure 4: Hyperparameter sensitivity in CAt-Walk. AUC on inductive test hyperedges are reported.

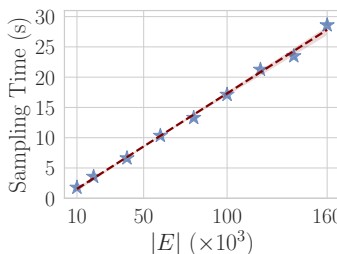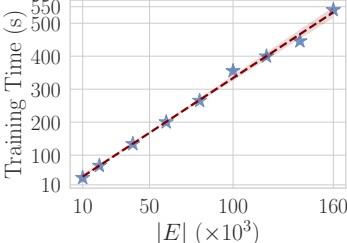

Figure 5: Scalability evaluation: The runtime of (**left**) SᴇᴛWᴀʟᴋ extraction and (**right**) the training time of CAᴛ-Wᴀʟᴋ over one epoch on High School (using different $|E|$ for training).

increasing the number of sampled SᴇᴛWᴀʟᴋs produces better performance. The main reason is that the model has more extracted structural and temporal information from the network. Also, notably, we observe that only a small number of sampled SᴇᴛWᴀʟᴋs are needed to achieve competitive performance: in the best case 1 and in the worst case 16 sampled SᴇᴛWᴀʟᴋs per each hyperedge are needed. The middle figure reports the effect of the length of SᴇᴛWᴀʟᴋs on performance. These results show that performance peaks at certain SᴇᴛWᴀʟᴋ lengths and the exact value varies with the dataset. That is, longer SᴇᴛWᴀʟᴋs are required for the networks that evolve according to more complicated laws encoded in temporal higher-order motifs. The right figure shows the effect of the temporal bias coefficient $\alpha$. Results suggest that $\alpha$ has a dataset-dependent optimal interval. That is, a small $\alpha$ suggests an almost uniform sampling of interaction history, which results in poor performance when the short-term dependencies (interactions) are more important in the dataset. Also, very large $\alpha$ might damage performance as it makes the model focus on the most recent few interactions, missing long-term patterns.

**Scalability Analysis.** In this part, we investigate the scalability of CAᴛ-Wᴀʟᴋ. To this end, we use different versions of the High School dataset with different numbers of hyperedges from $10^4$ to $1.6 \times 10^5$. Figure 5 (left) reports the runtimes of SᴇᴛWᴀʟᴋ sampling and Figure 5 (right) reports the runtimes of CAᴛ-Wᴀʟᴋ for training one epoch using $M = 8$, $m = 3$ with batch-size = 64. Interestingly, our method scales linearly with the number of hyperedges, which enables it to be used on long hyperedge streams and large hypergraphs.

## 5   Conclusion, Limitation, and Future Work

We present CAᴛ-Wᴀʟᴋ, an inductive hypergraph representation learning that learns both higher-order patterns and the underlying dynamic laws of temporal hypergraphs. CAᴛ-Wᴀʟᴋ uses SᴇᴛWᴀʟᴋs, a new temporal, higher-order random walk on hypergraphs that are provably more expressive than existing walks on hypergraphs, to extract temporal higher-order motifs from hypergraphs. CAᴛ-Wᴀʟᴋ then uses a two-step, set-based anonymization process to establish the correlation between the extracted motifs. We further design a permutation invariant pooling strategy, SᴇᴛMɪxᴇʀ, for aggregating nodes' positional encodings in a hyperedge to obtain hyperedge level positional encodings. Consequently, the experimental results show that CAᴛ-Wᴀʟᴋ ① produces superior performance compared to the state-of-the-art in temporal hyperedge prediction tasks, and ② competitive performance in temporal node classification tasks. These results suggest many interesting directions for future studies: Using CAᴛ-Wᴀʟᴋ as a positional encoder in existing anomaly detection frameworks to design an inductive anomaly detection method on hypergraphs. There are, however, a few limitations: Currently, CAᴛ-Wᴀʟᴋ uses *fixed-length* SᴇᴛWᴀʟᴋs, which might cause suboptimal performance. Developing a procedure to learn from SetWalks of varying lengths might produce better results.

## Acknowledgments and Disclosure of Funding

We acknowledge the support of the Natural Sciences and Engineering Research Council of Canada (NSERC).
Nous remercions le Conseil de recherches en sciences naturelles et en génie du Canada (CRSNG) de son soutien.

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

# Appendix

## Table of Contents

# A    Preliminaries, Backgrounds, and Motivations

We begin by reviewing the preliminaries and background concepts that we refer to in the main paper. Next, we discuss the fundamental differences between our method and techniques from prior work.

## A.1    Anonymous Random Walks

Micali and Zhu [45] studied Anonymous Walks (AWs), which replace a *node's identity* by the order of its appearance in each walk. Given a simple network, an AW starts from a node, performs random walks over the graph to collect a sequence of nodes, $W : (u_1, u_2, \ldots, u_k)$, and then replaces the node identities by their order of appearance in each walk. That is:

$$\text{ID}_{\text{AW}}(w_0, W) = |\{u_1, u_2, \ldots, u_{k^*}\}| \text{ where } k^* \text{ is the smallest index such that } u_{k^*} = w_0. \quad (10)$$

While this method is a simple anonymization process, it misses the correlation between different walks and assigns new node identities based on only one single walk. The correlation between different walks is more important in temporal networks to assign new node identities, as a single walk cannot capture the frequency of a pattern over time [33]. To this end, Wang et al. [33] design a set-based anonymization process that assigns new node identities based on a set of sampled walks. Given a vertex $u$, they sample $M$ walks with length $m$ starting from $u$ and store them in $S_u$. Next, for each node $w_0$ that appears on at least one walk in $S_u$, they assign a vector to each node as its hidden identity [33]:

$$g(w_0, S_u)[i] = |\{W | W \in S_u, W[i] = w_0\}| \quad \forall i \in \{0, \ldots, m\}, \quad (11)$$

where $W[i]$ shows the $i$-th node in the walk $W$. This anonymization process not only hides the identity of vertices but it also can establish such hidden identity based on different sampled walks, capturing the correlation between several walks starting from a vertex.

Both of these anonymization processes are designed for graphs with pair-wise interactions, and there are three main challenges in adopting them for hypergraphs: ① To capture higher-order patterns, we use SETWALKS, which are a sequence of hyperedges. Accordingly, we need an encoding for the position of hyperedges. A natural attempt to encode the position of hyperedges is to count the position of hyperedges across sampled SETWALKS, as CAW [33] does for nodes. However, this approach misses the similarity of hyperedges with many nodes in common. That is, given two hyperedges $e_1 = \{u_1, u_2, \ldots, u_k\}$ and $e_2 = \{u_1, u_2, \ldots, u_k, u_{k+1}\}$. Although we want to encode the position of these two hyperedges, we also want these two hyperedges to have almost the same encoding as they share many vertices. Accordingly, we suggest viewing a hyperedge as a set of vertices. We first encode the position of vertices, and then we aggregate the position encodings of nodes that are connected by a hyperedge to compute the positional encoding of the hyperedge. ② However, since we focus on undirected hypergraphs, the order of a hyperedge's vertices in the aggregation process should not affect the hyperedge positional encodings. Therefore, we need a permutation invariant pooling strategy. ③ While several existing studies used simple pooling functions such as MEAN(.) or SUM(.) [26], these pooling functions do not capture the higher-order dependencies between obtained nodes' position encodings, missing the advantage of higher-order interactions. That is, a pooling function such as MEAN(.) is a non-parametric method that sees the positional encoding of each node in a hyperedge separately. Therefore, it is unable to aggregate them in a non-linear manner, which, depending on the data, can miss information. To address challenges ② and ③, we design SETMIXER, a permutation invariant pooling strategy that uses MLPs to learn how to aggregate positional encodings of vertices in a hyperedge to compute the hyperedge positional encoding.

## A.2    Random Walk on Hypergraphs

Chung [98] presents some of the earliest research on the hypergraph Laplacian, defining the Laplacian of the $k$-uniform hypergraph. Following this line of work, Zhou et al. [18] defined a two-step CE-based random walk-based Laplacian for general hypergraphs. Given a node $u$, in the first step, we uniformly sample a hyperedge $e$ including node $u$, and in the second step, we uniformly sample a node in $e$. Following this idea, several studies developed more sophisticated (weighted) CE-based random walks on hypergraphs [20]. However, Chitra and Raphael [40] shows that random walks on hypergraphs with edge-independent node weights are limited to capturing pair-wise interactions, making them unable to capture higher-order information. To address this limitation, they designed an edge-dependent sampling procedure of random walks on hypergraphs. Carletti et al. [37] and Carletti

et al. [99] argued that to sample more informative walks from a hypergraph, we must consider the degree of hyperedges in measuring the importance of vertices in the first step. Concurrently, some studies discuss the dependencies among hyperedges and define the *s*-th Laplacian based on simple walks on the dual hypergraphs [41, 100]. Finally, more sophisticated random walks with non-linear Laplacians have been designed [23, 101–103].

SETWALKS addresses three main drawbacks from existing methods: ① None of these methods are designed for temporal hypergraphsm so they cannot capture temporal properties of the network. Also, natural attempts to extend them to temporal hypergraphs and let the walker uniformly walk over time ignore the fact that recent hyperedges are more informative than older ones (see Table 2). To address this issue, SETWALK uses a temporal bias factor in its sampling procedure (Equation 1). ② Existing hypergraph random walks are unable to capture either higher-order interactions of vertices or higher-order dependencies of hyperedges. That is, random walks with edge-independent weights [37] are not able to capture higher-order interactions and are equivalent to simple random walks on the CE of the hypergraph [40]. The expressivity of random walks on hypergraphs with edge-dependent walks is also limited when we have a limited number of sampled walks (see Theorem 1). Finally, defining a hypergraph random walk as a random walk on the dual hypergraph also cannot capture the higher-order dependencies of hyperedges (see Appendix C and Appendix D). SETWALK by its nature is able to walk over hyperedges (instead of vertices) and time and can capture higher-order interactions. Also, with a structural bias factor in its sampling procedure, which is based on hyperedge-dependent node weights, it is more informative than a simple random walk on the dual hypergraph, capturing higher-order dependencies of hyperedges. See Appendix C for further discussion.

## A.3   MLP-Mixer

MLP-MIXER [44] is a family of models, based on multi-layer perceptions (MLPs), that are simple, amenable to efficient implementation, and robust to long-term dependencies (unlike RNNs, attention mechanisms, and Transformers [83]) with a wide array of applications from computer vision [104] to neuroscience [105]. The original architecture is designed for image data, where it takes image tokens as inputs. It then encodes them with a linear layer, which is equivalent to a convolutional layer over the image tokens, and updates their representations with a sequence of feed-forward layers applied to image tokens and features. Accordingly, we can divide the architecture of MLP-MIXER into two main parts: ① Token Mixer: The main intuition of the token mixer is to clearly separate the cross-location operations and learn the cross-feature (cross-location) dependencies. ② Channel Mixer: The intuition behind the channel mixer is to clearly separate the per-location operations and provide positional invariance, a prominent feature of convolutions. In both MIXER and SETMIXER we use the channel mixer as designed in MLP-MIXER. Next, we discuss the token mixer and its limitation in mixing features in a permutation variant manner:

**Token Mixer.** Let $\mathbf{E}$ be the input of the MLP-MIXER, then the token mixer phase is defined as:

$$\mathbf{H}_{\text{token}} = \mathbf{E} + \mathbf{W}_{\text{token}}^{(2)} \sigma \left( \mathbf{W}_{\text{token}}^{(1)} \texttt{LayerNorm}(\mathbf{E})^T \right)^T, \qquad (12)$$

where $\sigma(.)$ is nonlinear activation function (usually GeLU [80]). Since it feeds the input's columns to an MLP, it mixes the cross-feature information, which results in the MLP-MIXER being sensitive to permutation. Although natural attempts to remove the token mixer or its linear layer can produce a permutation invariant method, it misses cross-feature dependencies, which are the main motivation for using the MLP-MIXER architecture. To address this issue, SETMIXER uses the `Softmax`(.) function over features. Using `Softmax` over features can be seen as cross-feature normalization, which can capture their dependencies. While `Softmax`(.) is a non-parametric method that can bind token-wise information, it is also permutation equivariant, and as we prove in Appendix E.3, makes the SETMIXER permutation invariant.

# B   Additional Related Work

## B.1   Learning (Multi)Set Functions

(Multi)set functions are pooling architectures for (multi)sets with a wide array of applications in many real-world problems including few-shot image classification [106], conditional regression [107], and causality discovery [108]. Zaheer et al. [97] develop DEEPSETS, a universal approach to parameterize

the (multi)set functions. Following this direction, some works design attention mechanisms to learn multiset functions [109], which also inspired Baek et al. [110] to adopt attention mechanisms designed for (multi)set functions in graph representation learning. Finally, Chien et al. [25] build the connection between learning (multi)set functions with propagations on hypergraphs. To the best of our knowledge, SETMIXER is the first adaptive permutation invariant pooling strategy for hypergraphs, which views each hyperedge as a set of vertices and aggregates node encodings by considering their higher-order dependencies.

### B.2 Simplicial Complexes Representation Learning

Simplicial complexes can be considered a special case of hypergraphs and are defined as a collection of polytopes such as triangles and tetrahedra, which are called simplices [111]. While these frameworks can be used to represent higher-order relations, simplicial complexes require the downward closure property [112]. That is, every substructure or face of a simplex contained in a complex $\mathcal{K}$ is also in $\mathcal{K}$. Recently, to encode higher-order interactions, representation learning on simplicial complexes has attracted much attention [6, 113–119]. The first group of methods extend node2vec [67] to simplicial complexes with random walks on interactions through Hasse diagrams and simplex connections inside $p$-chains [113, 115]. With the recent advances in message-passing-based methods, several studies focus on designing neural networks on simplicial complexes [116–119]. Ebli et al. [116] introduced Simplicial neural networks (SNN), a generalization of spectral graph convolutio,n to simplicial complexes with higher-order Laplacian matrices. Following this direction, some works propose simplicial convolutional neural networks with different simplicial filters to exploit the relationships in upper- and lower-neighborhoods [117, 118]. Finally, the last group of studies use an encoder-decoder architecture as well as message-passing to learn the representation of simplicial complexes [114, 120].

CAT-WALK is different from all these methods in three main aspects: ① Contrary to these methods, CAT-WALK is designed for temporal hypergraphs and is capable of capturing higher-order temporal properties in a streaming manner, avoiding the drawbacks of snapshot-based methods. ② CAT-WALK works in the inductive setting by extracting underlying dynamic laws of the hypergraph, making it generalizable to unseen patterns and nodes. ③ All these methods are designed for simplicial complexes, which are special cases of hypergraphs, while CAT-WALK is designed for general hypergraphs and does not require any assumption of the downward closure property.

### B.3 How Does CAT-WALK Differ from Existing Works? (Contributions)

As we discussed in Appendix A.2, existing random walks on hypergraphs are unable to capture either ① higher-order interactions between nodes or ② higher-order dependencies of hyperedges. Moreover, all these walks are for static hypergraphs and are not able to capture temporal properties. To this end, we design SETWALK a higher-order temporal walk on hypergraphs. Naturally, SETWALKS are capable of capturing higher-order patterns as a SETWALK is defined as a sequence of hyperedges. We further design a new sampling procedure with temporal and structural biases, making SETWALKS capable of capturing higher-order dependencies of hyperedges. To take advantage of complex information provided by SETWALKS as well as training the model in an inductive manner, we design a two-step anonymization process with a novel pooling strategy, called SETMIXER. The anonymization process starts with encoding the position of vertices with respect to a set of sampled SETWALKS and then aggregates node positional encodings via a non-linear permutation invariant pooling function, SETMIXER, to compute their corresponding hyperedge positional encodings. This two-step process lets us capture structural properties while we also care about the similarity of hyperedges. Finally, to take advantage of continuous-time dynamics in data and avoid the limitations of sequential encoding, we design a neural network for temporal walk encoding that leverages a time encoding module to encode time as well as a MIXER module to encode the structure of the walk.

## C SETWALK and Random Walk on Hypergraphs

We reviewed existing random walks in Appendix A.2. Here, we discuss how these concepts are different from SETWALKS and investigate whether SETWALKS are more expressive than these methods.

As we discussed in Sections 1 and 3.2, there are two main challenges for designing random walks on hypergraphs: ① Random walks are a sequence of *pair-wise* interconnected vertices, even though

edges in a hypergraph connect *sets* of vertices. ② A sampling probability of a walk on a hypergraph must be different from its sampling probability on the CE of the hypergraph [37–43]. To address these challenges, most existing works on random walks on hypergraphs ignore ① and focus on ② to distinguish the walks on simple graphs and hypergraphs, and ① is relatively unexplored. To this end, we answer the following questions:

**Q1:  Can ② alone be sufficient to take advantage of higher-order interactions?** First, semantically, decomposing hyperedges into sequences of simple pair-wise interactions (CE) loses the semantic meaning of the hyperedges. Consider the collaboration network in Figure 1. When decomposing the hyperedges into pair-wise interactions, both $(A, B, C)$ and $(H, G, E)$ have the same structure (a triangle), while the semantics of these two structures in the data are completely different. That is, $(A, B, C)$ have *all* published a paper together, while each pair of $(H, G, E)$ separately have published a paper. One might argue that although the output of hypergraph random walks and simple random walks on the CE might be the same, the sampling probability of each walk is different and with a large number of samples, our model can distinguish these two structures. In Theorem 1 (proof in Appendix E) we theoretically show that when we have a finite number of hypergraph walk samples, $M$, there is a hypergraph $\mathcal{G}$ such that with $M$ hypergraph walks, the $\mathcal{G}$ and its CE are not distinguishable. Note that in reality, the bottleneck for the number of sampled walks in machine learning-based methods is memory. Accordingly, even with tuning the number of samples for each dataset, the size of samples is bounded by a small number. This theorem shows that with a limited budget for walk sampling, ② alone is not enough to capture higher-order patterns.

**Q2:  Can addressing ① alone be sufficient to take advantage of higher-order interactions?** To answer this question, we use the extended version of the edge-to-vertex dual graph concept for hypergraphs:

**Definition 4** (Dual Hypergraph). *Given a hypergraph $\mathcal{G} = (\mathcal{V}, \mathcal{E})$, the dual hypergraph of $\mathcal{G}$ is defined as $\tilde{\mathcal{G}} = (\tilde{\mathcal{V}}, \tilde{\mathcal{E}})$, where $\tilde{\mathcal{V}} = \mathcal{E}$ and a hyperedge $\tilde{e} = \{e_1, e_2, \ldots, e_k\} \in \tilde{\mathcal{E}}$ shows that $\bigcap_{i=1}^{k} e_i \neq \emptyset$.*

To address ①, we need to see walks on hypergraphs as a sequence of hyperedges (instead of a sequence of pair-wise connected nodes). One can interpret this as a hypergraph walk on the dual hypergraph. That is, each hypergraph walk on the dual graph is a sequence of $\mathcal{G}$'s hyperedges: $e_1 \to e_2 \to \cdots \to e_k$. However, as shown by Chitra and Raphael [40], each walk on hypergraphs with edge-independent weights for sampling vertices is equivalent to a simple walk on the (weighted) CE graph. To this end, addressing ② alone can be equivalent to sample walks on the CE of the dual hypergraph, which misses the higher-order interdependencies of hyperedges and their intersections.

Based on the above discussion, both ① and ② are required to capture higher-order interaction between nodes as well as higher-order interdependencies of hyperedges. The definition of SETWALKS (Definition 3) with *structural bias*, introduced in Equation 1, satisfies both ① and ②. In the next section, we discuss how a simple extension of SETWALKS can not only be more expressive than all existing walks on hypergraphs and their CEs, but its definition is universal, and all these methods are special cases of extended SETWALK.

## C.1   Extension of SETWALKS

Random walks on hypergraphs are simple but less expressive methods for extracting network motifs, while SETWALKS are more complex patterns that provide more expressive motif extraction approaches. One can model the trade-off of simplicity and expressivity to connect all these concepts in a single notion of walks. To establish a connection between SETWALKS and existing walks on hypergraphs, as well as a universal random walk model on hypergraphs, we extend SETWALKS to $r$-SETWALKS, where parameter $r$ controls the size of hyperedges that appear in the walk:

**Definition 5** ($r$-SETWALK). *Given a temporal hypergraph $\mathcal{G} = (\mathcal{V}, \mathcal{E}, X)$, and a threshold $r \in \mathbb{Z}^+$, a $r$-SETWALK with length $\ell$ on temporal hypergraph $\mathcal{G}$ is a randomly generated sequence of hyperedges (sets):*

$$\text{Sw} : (e_1, t_{e_1}) \to (e_2, t_{e_2}) \to \cdots \to (e_\ell, t_{e_\ell}),$$

*where $e_i \in \mathcal{E}$, $|e_i| \leq r$, $t_{e_{i+1}} < t_{e_i}$, and the intersection of $e_i$ and $e_{i+1}$ is not empty, $e_i \cap e_{i+1} \neq \emptyset$. In other words, for each $1 \leq i \leq \ell - 1$: $e_{i+1} \in \mathcal{E}^{t_i}(e_i)$. We use $\text{Sw}[i]$ to denote the i-th hyperedge-time pair in the SETWALK. That is, $\text{Sw}[i][0] = e_i$ and $\text{Sw}[i][1] = t_{e_i}$.*

The only difference between this definition and Definition 3 is that $r$-SETWALK limits hyperedges in the walk to hyperedges with size at most $r$. The sampling process of $r$-SETWALKs is the same as that of SETWALK (introduced in Section 3.2 and Appendix D), while we only sample hyperedges with size at most $r$. Now to establish the connection of $r$-SETWALKs and existing walks on hypergraphs, we define the extended version of the clique expansion technique:

**Definition 6** ($r$-Projected Hypergraph). *Given a hypergraph $\mathcal{G} = (\mathcal{V}, \mathcal{E})$ and an integer $r \geq 2$, we construct the (weighted) $r$-projected hypergraph of $\mathcal{G}$ as a hypergraph $\hat{\mathcal{G}}_r = (\mathcal{V}, \hat{\mathcal{E}}_r)$, where for each $e = \{u_1, u_2, \ldots, u_k\} \in \mathcal{E}$:*

   *1. if $k \leq r$: add $e$ to the $\hat{\mathcal{E}}_r$,*

   *2. if $k \geq r + 1$: add $e_i = \{u_{i_1}, u_{i_2}, \ldots, u_{i_r}\}$ to $\hat{\mathcal{E}}_r$, for every possible $\{i_1, i_2, \ldots, i_r\} \subseteq \{1, 2, \ldots, k\}$.*

Each of steps 1 or 2 can be done in a weighted manner. In other words, we approximate each hyperedge with a size of more than $r$ with $\binom{k}{r}$ (weighted) hyperedges with size $r$. For example, when $r = 2$, the 2-projected graph of $\mathcal{G}$ is equivalent to its clique expansion, and $r = \infty$ is the hypergraph itself. Furthermore, we define the Union Projected Hyperaph (UP hypergraph) as the union of all $r$-projected hypergraphs, i.e., $\mathcal{G}^* = (\mathcal{V}, \bigcup_{r=2}^{\infty} \hat{\mathcal{E}}_r)$. Note that the UP hypergraph has the downward closure property and is equivalent to the simplicial complex representation of the hypergraph $\mathcal{G}$. The next proposition establishes the universality of the $r$-SETWALK concept.

**Proposition 2.** *Edge-independent random walks on hypergraphs [37], edge-dependent random walks on hypergraphs [40], and simple random walks on the CE of hypergraphs are all special cases of $r$-SETWALK, when applied to the 2-projected graph, UP hypergraph, and 2-projected graph, respectively. Furthermore, all the above methods are less expressive than $r$-SETWALKs.*

The proof of this proposition is in Appendix E.5.

## D   Efficient Hyperedge Sampling

For sampling SETWALKs, inspired by Wang et al. [33], we use two steps: ① Online score computation: we assign a set of scores to each incoming hyperedge. ② Iterative sampling: we use assigned scores in the previous step to sample hyperedges in a SETWALK.

---
**Algorithm 1** Online Score Computation

---
**Input:** Given a hypergraph $\mathcal{G} = (\mathcal{V}, \mathcal{E})$ and $\alpha \in [0, 1]$
**Output:** A probability score for each vertex
 1: $P \leftarrow \emptyset$;
 2: **for** $(e, t) \in \mathcal{E}$ with an increasing order of $t$ **do**
 3:     $P_{e,t}[0] \leftarrow \exp(\alpha t)$;
 4:     $P_{e,t}[1] \leftarrow 0$, $P_{e,t}[2] \leftarrow 0$, $P_{e,t}[3] \leftarrow \emptyset$;
 5:     **for** $u \in e$ **do**
 6:         **for** $e_n \in \mathcal{E}^t(u)$ **do**
 7:             **if** $e_n$ is not visited **then**
 8:                 $P_{e,t}[2] \leftarrow P_{e,t}[2] + \exp(\varphi(e_n, e))$;
 9:                 $P_{e,t}[3] \leftarrow P_{e,t}[3] \cup \{\exp(\varphi(e_n, e))\}$;
10:                 $P_{e,t}[1] \leftarrow P_{e,t}[1] + \exp(\alpha \times t_n)$;
     **return** $P$;

---

**Online Score Computation.** The first part essentially works in an online manner and assigns each new incoming hyperedge $e$ a four-tuple of scores:

$$P_{e,t}[0] = \exp(\alpha \times t), \qquad\qquad P_{e,t}[1] = \sum_{(e',t') \in \mathcal{E}^t(e)} \exp(\alpha \times t')$$

$$P_{e,t}[2] = \sum_{(e',t') \in \mathcal{E}^t(e)} \exp(\varphi(e, e')), \qquad\qquad P_{e,t}[3] = \{\exp(\varphi(e, e'))\}_{(e',t') \in \mathcal{E}^t(e)}$$

**Iterative Sampling.** In the iterative sampling algorithm, we use pre-computed scores by Algorithm 1 and sample a hyperedge $(e, t)$ given a previously sampled hyperedge $(e_p, t_p)$. In the next proposition,

**Algorithm 2** Iterative SETWALK Sampling

---

**Input:** Given a hypergraph $\mathcal{G} = (\mathcal{V}, \mathcal{E})$, $\alpha \in [0, 1]$, and previously sampled hyperedge $(e_p, t_p)$
**Output:** Next sampled hyperedge $(e, t)$
1: **for** $(e, t) \in \mathcal{E}^{t_p}(e_p)$ with an decreasing order of $t$ **do**
2:     Sample $b \sim$ UNIFORM$(0, 1)$;
3:     Get $P_{e,t}[0]$, $P_{e_p,t_p}[1]$, $P_{e_p,t_p}[2]$ and $\varphi(e, e_p)$ from the output of Algorithm 1;
4:     $\mathcal{P} \leftarrow$ Normalize $\frac{P_{e,t}[0]}{P_{e_p,t_p}[1]} \times \frac{\exp(\varphi(e,e_p))}{P_{e_p,t_p}[2]}$;
5:     **if** $b < \mathcal{P}$ **then return** $(e, t)$;
    **return** $(e_X, t_X)$;         ▷ $(e_X, t_X)$ is a dummy empty hyperedge signaling the end of algorithm.

---

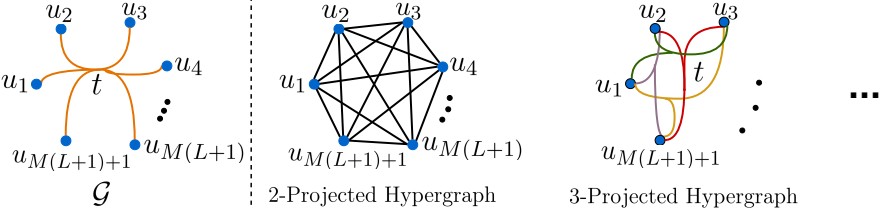

Figure 6: The example of a hypergraph $\mathcal{G}$ and its 2- and 3-projected hypergraphs.

we show that this sampling algorithm samples each hyperedge with the probability mentioned in Section 3.2.

**Proposition 3.** *Algorithm 2 sample a hyperedge $(e, t)$ after $(e_p, t_p)$ with a probability proportional to $\mathbb{P}[(e, t)|(e_p, t_p)]$ (Equation 1).*

**How can this sampling procedure capture higher-order patterns?** As discussed in Appendix C, SETWALKs on $\mathcal{G}$ can be interpreted as a random walk on the dual hypergraph of $\mathcal{G}$, $\tilde{\mathcal{G}}$. However, a simple (or hyperedge-independent) random walk on the dual hypergraph is equivalent to the walk on the CE of the dual hypergraph [40, 41], missing the higher-order dependencies of hyperedges. Inspired by Chitra and Raphael [40], we use hyperedge-dependent weights $\Gamma : \mathcal{V} \times \mathcal{E} \to \mathbb{R}^{\geq 0}$ and sample hyperedges with a probability proportional to $\exp\left(\sum_{u \in e \cap e_p} \Gamma(u, e)\Gamma(u, e')\right)$, where $e_p$ is the previously sampled hyperedge. In the dual hypergraph $\tilde{\mathcal{G}} = (\mathcal{E}, \mathcal{V})$, we assign a score $\tilde{\Gamma} : \mathcal{E} \times \mathcal{V} \to \mathbb{R}^{\geq 0}$ to each pair of $(e, u)$ as $\tilde{\Gamma}(e, u) = \Gamma(u, e)$. Now, a SETWALK with this sampling procedure is equivalent to the edge-dependent hypergraph walk on the dual hypergraph of $\mathcal{G}$ with edge-dependent weight $\tilde{\Gamma}(.)$. Chitra and Raphael [40] show that an edge-dependent hypergraph random walk can capture some information about higher-order interactions and is not equivalent to a simple walk on the weighted CE of the hypergraph. Accordingly, even on the dual hypergraph, SETWALK with this sampling procedure can capture higher-order dependencies of hyperedges and is not equivalent to a simple walk on the CE of the dual hypergraph $\tilde{\mathcal{G}}$. We conclude that, unlike existing random walks on hypergraphs [37, 38, 41, 79], SETWALK can capture both higher-order interactions of nodes, and, based on its sampling procedure, higher-order dependencies of hyperedges.

## E Theoretical Results

### E.1 Proof of Theorem 1

**Theorem 1.** *A random SETWALK is equivalent to neither the hypergraph random walk, the random walk on the CE graph, nor the random walk on the SE graph. Also, for a finite number of samples of each, SETWALK is more expressive than existing walks.*

*Proof.* In this proof, we focus on the hypergraph random walk and simple random walk on the CE. The proof for the SE graph is the same and also it has been proven that the SE graph and the CE of a hypergraph have close (or equal in uniform hypergraphs) Laplacian and have the same expressiveness power in the representation of hypergraphs [121–123].

First, note that each SETWALK can be approximately decomposed to a set of either hypergraph walks, simple random walks, or walk on the SE. Moreover, each of these walks can be mapped to

a corresponding SETWALK (but not a bijective mapping), by sampling hyperpedges corresponding to each consecutive pair of nodes in these walks. Accordingly, SETWALKS includes the information provided by these walks and so its expressiveness is not less than these methods. To this end, next, we discuss two examples in two different tasks for which SETWALKS are successful while other walks fail.

① In the first task, we want to see if there exists a pair of hypergraphs with different semantics that SETWALKS can distinguish, but other walks cannot. We construct such hypergraphs. Let $\mathcal{G} = (\mathcal{V}, \mathcal{E})$ be a hypergraph with $\mathcal{V} = \{u_1, u_2, \ldots, u_N\}$ and $\mathcal{E} = \{(e, t_i)\}_{i=1}^T$, where $e = \{u_1, u_2, \ldots, u_N\}$ and $t_1 < t_2 < \cdots < t_T$. Also, let $\mathcal{A}$ be an edge-independent hypergraph random walk (or random walk on the CE) sampling algorithm. Chitra and Raphael [40] show that each of these walks is equivalent to a random walk on the weighted CE. Assume that $\xi(.)$ is a function that assigns weights to edges in $\mathcal{G}^* = (\mathcal{V}, \mathcal{E}^*)$, the weighted CE of the $\mathcal{G}$, such that a hypergraph random walk on $\mathcal{G}$ is equivalent to a walk on this weighted CE graph. Next, we construct a weighted hypergraph $\mathcal{G}' = (\mathcal{V}, \mathcal{E}')$ with the same set of vertices but with $\mathcal{E}' = \bigcup_{k=1}^T \{((u_i, u_j), t_k)\}_{u_i, u_j \in \mathcal{V}}$, such that each edge $e_{i,j} = (u_i, u_j)$ is associated with a weight $\xi(e_{i,j})$. Clearly, sampling procedure $\mathcal{A}$ on $\mathcal{G}$ and $\mathcal{G}'$ are the same, while they have different semantics. For example, assume that both are collaboration networks. In $\mathcal{G}$, all vertices have published a single paper together, while in $\mathcal{G}'$, each pair of vertices have published a separate paper together. The proof for the hypergraph random walk with hyperedge-dependent weights is the same, while we construct weights of the hypergraph $\mathcal{G}'$ based on the sampling probability of hyperedges in the hypergraph random walk procedure.

② Next, in the second task, we investigate the expressiveness of these walks for reconstructing hyperedges. That is, we want to see that given a perfect classifier, can these walks provide enough information to detect higher-order patterns in the network. To this end, we show that for a finite number of samples of each walk, SETWALK is more expressive than all of these walks in detecting higher-order patterns. Let $M$ be the maximum number of samples and $L$ be the maximum length of walks, we show that for any $M \geq 2$ and $L \geq 2$ there exists a pair of hypergraphs $\mathcal{G}$, with higher-order interactions, and $\mathcal{G}'$, with pairwise interactions, such that SETWALKS can distinguish them, while they are indistinguishable by any of these walks. We construct a temporal hypergraph $\mathcal{G} = (\mathcal{V}, \mathcal{E})$ as a hypergraph with $\mathcal{V} = \{u_1, u_2, \ldots, u_{M(L+1)+1}\}$ and $\mathcal{E} = \{(e, t_i)\}_{i=1}^L$, where $e = \{u_1, u_2, \ldots, u_{M(L+1)+1}\}$ and $t_1 < t_2 < \cdots < t_L$. We further construct $\mathcal{G}' = (\mathcal{V}, \mathcal{E}')$ with the same set of vertices but with $\mathcal{E}' = \bigcup_{k=1}^L \{((u_i, u_j), t_k)\}_{u_i, u_j \in \mathcal{V}}$. Figure 6 illustrates $\mathcal{G}$ and its projected graphs at a given timestamp $t \in \{t_1, t_2, \ldots, t_L\}$.

SETWALK with only one sample, Sw, can distinguish interactions in these two hypergraphs. That is, let Sw : $(e, t_L) \rightarrow (e, t_{L-1}) \rightarrow \cdots \rightarrow (e, t_1)$ be the sample SETWALK from $\mathcal{G}$ (note that masking the time, this is the only SETWALK on $\mathcal{G}$, so in any case the sampled SETWALK is Sw). Since all interactions in $\mathcal{G}'$ are pairwise, *any* sampled SETWALK on $\mathcal{G}'$, Sw', includes only pairwise interactions, so Sw $\neq$ Sw', in any case. Accordingly, SETWALK can distinguish interactions in these two hypergraphs.

Since the output of hypergraph random walks, simple walks on the CE, and walks on the SE include only pairwise interactions, it seems that they are unable to detect higher-order patterns, so are unable to distinguish these two hypergraphs. However, one might argue that by having a large number of sampled walks and using a perfect classifier, which learned the distribution of sampled random walks and can detect whether a set of sampled walks is from a higher-order interaction, we might be able to detect higher-order interactions. To this end, we next assume that we have a perfect classifier $C(.)$ that can detect whether a set of sampled hypergraph walks, simple walks on the CE, or walks on the SE are sampled from a higher-order structure or pair-wise patterns. Next, we show that hypergraph random walks cannot provide enough information about every vertex for $C(.)$ to detect whether all vertices in $\mathcal{V}$ shape a hyperedge. To this end, assume that we sample $S = \{W_1, W_2, \ldots, W_M\}$ walks from hypergraph $\mathcal{G}$ and $S' = \{W_1', W_2', \ldots, W_M'\}$ walks from hypergraph $\mathcal{G}'$. In the best case scenario, since $C(.)$ is a perfect classifier, it can detect that $\mathcal{G}'$ includes only pair-wise interactions based on sampled walk $S'$. To distinguish these two hypergraphs, we need $C(.)$ to detect sampled walks from $\mathcal{G}$ (i.e., $S$) that come from a higher-order pattern. For any $M$ sampled walks with length $L$ from $\mathcal{G}$, we observe at most $M \times (L+1)$ vertices, so we have information about at most $M \times (L+1)$ vertices, unable to capture any information about the neighborhood of at least one vertex. Due to the symmetry of vertices, without loss of generality, we can assume that this vertex is $u_1$. This means that with these $M$ sampled hypergraph random walks with length $L$, we are not able to provide any information about node $u_1$ at any timestamp for $C(.)$. Therefore, even a perfect classifier $C(.)$ cannot verify whether $u_1$

is a part of higher-order interaction or pair-wise interaction, which completes the proof. Note that the proof for the simple random walk is completely the same.

$\square$

**Remark 1.** *Note that while the first task investigates the expressiveness of these methods with respect to their sampling procedure, the second tasks discuss the limitation and difference in their outputs.*

**Remark 2.** *Note that in reality, we can have neither an unlimited number of samples nor an unlimited walk length. Also, the upper bound for the number of samples or walk length depends on the RAM of the machine on which the model is being trained. In our experiments, we observe that usually, we cannot sample more than 125 walks with a batch size of 32.*

## E.2 Proof of Theorem 2

Before discussing the proof of Theorem 2 we first formally define what missing information means in this context.

**Definition 7** (Missing Information). *We say a pooling strategy like $\Psi(.)$ misses information if there is a model $\mathcal{M}$ such that using $\Psi(.)$ on top of the $\mathcal{M}$ (call it $\tilde{\mathcal{M}}$) decreases the expressive power of $\mathcal{M}$. That is, $\tilde{\mathcal{M}}$ has less expressive power than $\mathcal{M}$.*

**Theorem 2.** *Given an arbitrary positive integer $k \in \mathbb{Z}^+$, let $\Psi(.)$ be a pooling function such that for any set $S = \{w_1, \ldots, w_d\}$:*

$$\Psi(S) = \sum_{\substack{S' \subseteq S \\ |S'|=k}} f(S'), \tag{13}$$

*where $f$ is some function. Then the pooling function can cause missing information, limiting the expressiveness of the method to applying to the projected (hyper)graph of the hypergraph.*

*Proof.* The main intuition of this theorem is that a pooling function needs to capture higher-order dependencies of its input's elements and if it can be decomposed to a summation of functions that capture lower-order dependencies, it misses information. We show that, in the general case for a given $k \in \mathbb{Z}^+$, the pooling function $\Psi(.)$ when applied to a hypergraph $\mathcal{G}$ is at most as expressive as $\Psi(.)$ when applied to the $k$-projected hypergraph of $\mathcal{G}$ (Definition 6). Let $\mathcal{G} = (\mathcal{V}, \mathcal{E})$ be a hypergraph with $\mathcal{V} = \{u_1, u_2, \ldots, u_{k+1}\}$ and $\mathcal{E} = \{(\mathcal{V}, t)\} = \{((u_1, u_2, \ldots, u_{k+1}), t)\}$ for a given time $t$, and $\hat{\mathcal{G}} = (\mathcal{V}, \hat{\mathcal{E}})$ be its $k$-projected graph, i.e., $\hat{\mathcal{E}} = \{(e_1, t), \ldots, (e_{\binom{k+1}{k}}, t)\}$, where $e_i \subset \{u_1, u_2, \ldots, u_{k+1}\}$ such that $|e_i| = k$. Applying pooling function $\Psi(.)$ on the hypergraph $\mathcal{G}$ is equivalent to applying $\Psi(.)$ to the hyperedge $(\mathcal{V}, t) \in \mathcal{E}$, which provides $\Psi(\mathcal{V}) = \sum_{i=1}^{k+1} f(e_i)$. On the other hand, applying $\Psi(.)$ on projected graph $\hat{\mathcal{G}}$ means applying it on each hyperedge $e_i \in \hat{\mathcal{E}}$. Accordingly, since for each hyperedge $e_i \in \hat{\mathcal{E}}$ we have $\Psi(e_i) = f(e_i)$, all captured information by pooling function $\Psi(.)$ on $\hat{\mathcal{G}}$ is the set of $S = \{f(e_i)\}_{i=1}^{k+1}$. It is clear that $\Psi(\mathcal{V}) = \sum_{i=1}^{k+1} f(e_i)$ is less informative than $S = \{f(e_i)\}_{i=1}^{k+1}$ as it is the summation of elements in $S$ (in fact, $\Psi(\mathcal{V})$ cannot capture the non-linear combinations of positional encodings of vertices, while $S$ can). Accordingly, the provided information by applying $\Psi(.)$ on $\mathcal{G}$ cannot be more informative than applying $\Psi(.)$ on the $\mathcal{G}$'s $k$-projected hypergraph. $\square$

**Remark 3.** *Note that the pooling function $\Psi(.)$ is defined on a (hyper)graph and gets only (hyper)edges as input.*

**Remark 4.** *Although $\Psi(.) = \text{Mean}(.)$ cannot be written as Equation 13, we can simply see that the above proof works for this pooling function as well.*

## E.3 Proof of Theorem 3

**Theorem 3.** SetMixer *is permutation invariant and is a universal approximator of invariant multi-set functions. That is,* SetMixer *can approximate any invariant multi-set function.*

*Proof.* Let $\pi(S)$ be a given permutation of set $S$, we aim to show that $\Psi(S) = \Psi(\pi(S))$. We first recall the SetMixer and its two phases: Let $S = \{\mathbf{v}_1, \ldots, \mathbf{v}_d\}$, where $\mathbf{v}_i \in \mathbb{R}^{d_1}$, be the input set and $\mathbf{V} = [\mathbf{v}_1, \ldots, \mathbf{v}_d]^T \in \mathbb{R}^{d \times d_1}$ be its matrix representation:

$$\Psi(\mathbf{V}) = \text{Mean}\left(\mathbf{H}_{\text{token}} + \sigma\left(\texttt{LayerNorm}\left(\mathbf{H}_{\text{token}}\right)\mathbf{W}_s^{(1)}\right)\mathbf{W}_s^{(2)}\right), \quad \textit{(Channel Mixer)}$$

where

$$\mathbf{H}_{\text{token}} = \mathbf{V} + \sigma\left(\text{Softmax}\left(\text{LayerNorm}(\mathbf{V})^T\right)\right)^T. \qquad (\textit{Token Mixer})$$

Let $\pi(\mathbf{V}) = [\mathbf{v}_{\pi(1)}, \ldots, \mathbf{v}_{\pi(d)}]^T$ be a permutation of the input matrix $\mathbf{V}$. In the token mixer phase, None of LayerNorm, Softmax, and activation function $\sigma(.)$ can affect the order of elements (note that Softmax is applied row-wise). Accordingly, we can see the output of the token mixer is permuted by $\pi(.)$:

$$
\begin{aligned}
\mathbf{H}_{\text{token}}(\pi(\mathbf{V})) &= \pi(\mathbf{V}) + \sigma\left(\text{Softmax}\left(\text{LayerNorm}(\pi(\mathbf{V}))^T\right)\right)^T \\
&= \pi(\mathbf{V}) + \pi\left(\sigma\left(\text{Softmax}\left(\text{LayerNorm}(\mathbf{V})^T\right)\right)^T\right) \\
&= \pi\left(\mathbf{V} + \sigma\left(\text{Softmax}\left(\text{LayerNorm}(\mathbf{V})^T\right)\right)^T\right) \\
&= \pi\left(\mathbf{H}_{\text{token}}(\mathbf{V})\right).
\end{aligned}
\qquad (14)
$$

Next, in the channel mixer, by using Equation 14 we have:

$$
\begin{aligned}
\Psi(\pi(\mathbf{V})) &= \text{Mean}\left(\pi(\mathbf{H}_{\text{token}}) + \sigma\left(\text{LayerNorm}(\pi(\mathbf{H}_{\text{token}}))\,\mathbf{W}_s^{(1)}\right)\mathbf{W}_s^{(2)}\right) \\
&= \text{Mean}\left(\pi(\mathbf{H}_{\text{token}}) + \pi\left(\sigma\left(\text{LayerNorm}(\mathbf{H}_{\text{token}})\,\mathbf{W}_s^{(1)}\right)\right)\mathbf{W}_s^{(2)}\right) \\
&= \text{Mean}\left(\pi(\mathbf{H}_{\text{token}}) + \pi\left(\sigma\left(\text{LayerNorm}(\mathbf{H}_{\text{token}})\,\mathbf{W}_s^{(1)}\right)\mathbf{W}_s^{(2)}\right)\right) \\
&= \text{Mean}\left(\pi\left(\mathbf{H}_{\text{token}} + \mathbf{W}_s^{(2)}\sigma\left(\text{LayerNorm}(\mathbf{H}_{\text{token}})\,\mathbf{W}_s^{(1)}\right)\right)\right) \\
&= \Psi(\mathbf{V}).
\end{aligned}
\qquad (15)
$$

In the last step, we use the fact that Mean(.) is permutation invariant. Based on Equation 15 we can see that SetMixer is permutation invariant.

Since the token mixer is just normalization it is inevitable and cannot affect the expressive power of SetMixer. Also, channel mixer is a 2-layer MLP, which is the universal approximator of any function. Therefore, SetMixer is a universal approximator. □

### E.4 Proof of Theorem 4

**Theorem 4.** *The set-based anonymization method is more expressive than any existing anonymization strategies on the CE of the hypergraph. More precisely, there exists a pair of hypergraphs $\mathcal{G}_1 = (\mathcal{V}_1, \mathcal{E}_1)$ and $\mathcal{G}_2 = (\mathcal{V}_2, \mathcal{E}_2)$ with different structures (i.e., $\mathcal{G}_1 \ncong \mathcal{G}_2$) that are distinguishable by our anonymization process and are not distinguishable by the CE-based methods.*

*Proof.* To the best of our knowledge, there exist two anonymization processes for random walks by Wang et al. [33] and Micali and Zhu [45]. Both of these methods are designed for graphs and to adapt them to hypergraphs we need to apply them to the (weighted) CE. Here, we focus on the process designed by Wang et al. [33], which is more informative than the other. The proof for the Micali and Zhu [45] process is the same. Note that the goal of this theorem is to investigate whether a method can distinguish a hypergraph from its CE. Accordingly, this theorem does not provide any information about the expressivity of these methods in terms of the isomorphism test.

The proposed 2-step anonymization process can be seen as a positional encoding for both vertices and hyperedges. Accordingly, it is expected to assign different positional encodings to vertices and hyperedges of two non-isomorphism hypergraphs. To this end, we construct the same hypergraphs as in the proof of Theorem 1. Let $M$ be the number of sampled SetWalks with length $L$. We construct a temporal hypergraph $\mathcal{G} = (\mathcal{V}, \mathcal{E})$ as a hypergraph with $\mathcal{V} = \{u_1, u_2, \ldots, u_{M(L+1)+1}\}$ and $\mathcal{E} = \{(e, t_i)\}_{i=1}^L$, where $e = \{u_1, u_2, \ldots, u_{M(L+1)+1}\}$ and $t_1 < t_2 < \cdots < t_L$. We further construct $\mathcal{G}' = (\mathcal{V}, \mathcal{E}')$ with the same set of vertices but with $\mathcal{E}' = \bigcup_{k=1}^L \{((u_i, u_j), t_k)\}_{u_i, u_j \in \mathcal{V}}$. As we have seen in Theorem 1, random walks on the CE of the hypergraph cannot distinguish these two hypergraphs. Since CAW [33] also uses simple random walks, it cannot distinguish these two hypergraphs. Accordingly, after its anonymization process, it again cannot distinguish these two hypergraphs.

The main part of the proof is to show that in our method, the assigned positional encodings are different in these hypergraphs. The first step is to assign each node a positional encoding. Masking

the timestamps, there is only one SETWALK in the $\mathcal{G}$. Accordingly, the positional encodings of nodes in $\mathcal{G}$ are the same and non-zero. Given a SETWALK with length $L$ we might see at most $L \times (d_{\max} - 1) + 1$ nodes, where $d_{\max}$ is the maximum size of hyperedges in the hypergraph. Accordingly, with $M$ samples on $\mathcal{G}'$, which $d_{\max} = 2$, we can see at most $M \times (L + 1)$ vertices. Therefore, in any case, we assign a zero vector to at least one vertex. This proves that the positional encodings by SETWALKS are different in these two hypergraphs, and if the assigned hidden identities to counterpart nodes are different, clearly, feeding them to the SETMIXER results in different hyperedge encodings.

Note that each SETWALK can be decomposed into a set of causal anonymous walks [33]. Accordingly, it includes the information provided by these walks, so its expressiveness is not less than the CAW method on hypergraphs, which completes the proof of the theorem. □

Although the above statement completes the proof, next we discuss that even given the same positional encodings for vertices in these two hypergraphs, SETMIXER can capture higher-order interactions by capturing the size of the hyperedge. Recall token mixer phase in SETMIXER:

$$\mathbf{H}_{\text{token}} = \mathbf{V} + \sigma \left( \texttt{Softmax} \left( \texttt{LayerNorm}(\mathbf{V})^T \right) \right)^T,$$

where $\mathbf{V} = [\mathbf{v}_1, \ldots, \mathbf{v}_{M(L+1)+1}]^T \in \mathbb{R}^{(M(L+1)+1) \times d_1}$ and $\mathbf{v}_i \neq 0_{1 \times d_1}$ represents the positional encoding of $u_i$ in $\mathcal{G}$. We assumed that the positional encoding of $u_i$ in $\mathcal{G}'$ is the same. The input of the token mixer phase on $\mathcal{G}$ is $\mathcal{V}$ as all of them are connected by a hyperedge. Then we have:

$$(\mathbf{H}_{\text{token}})_{i,j} = \mathbf{v}_{i,j} + \sigma \left( \frac{\exp(\mathbf{v}_{i,j})}{\sum_{k=1}^{M(L+1)+1} \exp(\mathbf{v}_{k,j})} \right). \tag{16}$$

On the other hand, when applied to hypergraph $\mathcal{G}'$ and $(u_{k_1}, u_{k_2})$. We have:

$$(\mathbf{H}'_{\text{token}})_{i,j} = \mathbf{v}_{i,j} + \sigma \left( \frac{\exp(\mathbf{v}_{i,j})}{\exp(\mathbf{v}_{k_1,j}) + \exp(\mathbf{v}_{k_2,j})} \right), \quad i \in \{k_1, k_2\}. \tag{17}$$

Since we use zero padding, for any $i \geq 3$, $(\mathbf{H}_{\text{token}})_{i,j} \neq 0$ and $(\mathbf{H}'_{\text{token}})_{i,j} = 0$. These zero rows, which capture the size of the hyperedge, result in different encodings for each connection.

**Remark 5.** *To the best of our knowledge, the only anonymization process that is used on hypergraphs is by Liu et al. [78], which uses simple walks on the CE and is the same as Wang et al. [33]. Accordingly, it also suffers from the above limitation. Also, note that this theorem shows the limitation of these anonymization procedures when simply adopted to hypergraphs.*

### E.5 Proof of Proposition 2

**Proposition 2.** *Edge-independent random walks on hypergraphs [37], edge-dependent random walks on hypergraphs [40], and simple random walks on the CE of hypergraphs are all special cases of r-SETWALK, when applied to the 2-projected graph, UP graph, and 2-projected graph, respectively. Furthermore, all the above methods are less expressive than r-SETWALKs.*

*Proof.* For the first part, we discuss each walk separately:

① Simple random walks on the CE of the hypergraphs: We perform 2-SETWALKS on the (weighted) 2-projected hypergraph with $\Gamma(.) = 1$. Accordingly, for every two adjacent edges in the 2-Projected graph like $e$ and $e'$, we have $\varphi(e, e') = 1$. Therefore, it is equivalent to a simple random walk on the CE (2-projected graph).

② Edge-independent random walks on hypergraphs: As is shown by Chitra and Raphael [40], each edge-independent random walk on hypergraphs is equivalent to a simple random walk on the (weighted) CE of the hypergraph. Therefore, as discussed in ①, these walks are a special case of $r$-SETWALKs, when $r = 2$ and applied to (weighted) 2-Projected hypergraph.

③ Edge-dependent random walks on hypergraphs: Let $\Gamma'(e, u)$ be an edge-dependent weight function used in the hypergraph random walk sampling. For each node $u$ in the UP hypergraph, we store the set of $\Gamma'(e, u)$ that $e$ is a maximal hyperedge that $u$ belongs to. Note that there might be several maximal hyperedges that $u$ belongs to. Now, we perform 2-SETWALK sampling on the UP hypergraph with these weights and in each step, we sample each hyperedge with weight $\Gamma(u, e) = \Gamma'(e, u)$. It is

straightforward to show that given this procedure, the sampling probability of a hyperedge is the same in both cases. Therefore, edge-dependent random walks on hypergraphs are equivalent to 2-SETWALKS when applied to the UP hypergraph.

As discussed above, all these walks are special cases of $r$-SETWALKS and cannot be more expressive than $r$-SETWALKS. Also, as discussed in Theorem 1, all these walks are less expressive than SETWALKS, which are also special cases of $r$-SETWALKS, when $r = \infty$. Accordingly, all these methods are less expressive than $r$-SETWALKS. □

## F  Experimental Setup Details

### F.1  Datasets

We use 10 publicly available[3] benchmark datasets, whose descriptions are as follows:

- **NDC Class** [6]: The NDC Class dataset is a temporal higher-order network, in which each hyperedge corresponds to an individual drug, and the nodes contained within the hyperedges represent class labels assigned to these drugs. The timestamps, measured in days, indicate the initial market entry of each drug. Here, hyperedge prediction aims to predict future drugs.

- **NDC Substances** [6]: The NDC Substances is a temporal higher-order network, where each hyperedge represents an NDC code associated with a specific drug, while the nodes represent the constituent substances of the drug. The timestamps, measured in days, indicate the initial market entry of each drug. The hyperedge prediction task is the same as NDC Classes dataset.

- **High School** [6, 92]: The High School is a temporal higher-order network dataset constructed from interactions recorded by wearable sensors in a high school setting. The dataset captures a high school contact network, where each student/teacher is represented as a node and each hyperedge shows Face-to-face contact among individuals. Interactions were recorded at a resolution of 20 seconds, capturing all interactions that occurred within the previous 20 seconds. Node labels in this data are the class of students, and we focus on the node class "PSI" in our classification tasks.

- **Primary School** [6, 93]: The primary school dataset resembles the high school dataset, differing only in terms of the school level from which the data is collected. Node labels in this data are the class of students, and we focus on the node class "Teachers" in our classification tasks.

- **Congress Bill** [6, 94, 95]: Each node in this dataset represents a US Congressperson. Each hyperedge is a legislative bill in both the House of Representatives and the Senate, connecting the sponsors and co-sponsors of each respective bill. The timestamps, measured in days, indicate the date when each bill was introduced.

- **Email Enron** [6]: In this dataset nodes are email addresses at Enron and hyperedges are formed by emails, connecting the sender and recipients of each email. The timestamps have a resolution of milliseconds.

- **Email Eu** [6, 96]: In this dataset, the nodes represent email addresses associated with a European research institution. Each hyperedge consists of the sender and all recipients of the email. The timestamps in this dataset are measured with a resolution of 1 second.

- **Question Tags M (Math sx)** [6]: This dataset consists of nodes representing tags and hyperedges representing sets of tags applied to questions on math.stackexchange.com. The timestamps in the dataset are recorded at millisecond resolution and have been normalized to start at 0.

- **Question Tags U (Ask Ubuntu)** [6]: In this dataset, the nodes represent tags, and the hyperedges represent sets of tags applied to questions on askubuntu.com. The timestamps in the dataset are recorded with millisecond resolution and have been normalized to start at 0.

---

[3]https://www.cs.cornell.edu/~arb/data/

Table 3: Dataset statistics. HeP: **H**yp**e**redge **P**rediction, NC: **N**ode **C**lassification

| Dataset | NDC Class | High School | Primary School | Congress Bill | Email Enron | Email Eu | Question Tags M | Users-Threads | NDC Substances | Question Tags U |
|---|---|---|---|---|---|---|---|---|---|---|
| $|\mathcal{V}|$ | 1,161 | 327 | 242 | 1,718 | 143 | 998 | 1,629 | 125,602 | 5,311 | 3,029 |
| $|\mathcal{E}|$ | 49,724 | 172,035 | 106,879 | 260,851 | 10,883 | 234,760 | 822,059 | 192,947 | 112,405 | 271,233 |
| #Timestamps | 5,891 | 7,375 | 3,100 | 5,936 | 10,788 | 232,816 | 822,054 | 189,917 | 7,734 | 271,233 |
| Task | HeP | HeP& NC | HeP & NC | HeP | HeP | HeP | HeP | HeP | HeP | HeP |

- **Users-Threads** [6]: In this dataset, the nodes represent users on askubuntu.com, and a hyperedge is formed by users participating in a thread that lasts for a maximum duration of 24 hours. The timestamps in the dataset denote the time of each post, measured in milliseconds but normalized such that the earliest post begins at 0.

The statistics of these datasets can be found in Table 3.

## F.2 Baselines

We compare our method to eight previous state-of-the-art methods and baselines on the hyperedge prediction task:

- **CHESHIRE** [11]: Chebyshev spectral hyperlink predictor (CHESHIRE), is a hyperedge prediction methods that initializes node embeddings by directly passing the incidence matrix through a one-layer neural network. CHESHIRE treats a hyperedge as a fully connected graph (clique) and uses a Chebyshev spectral GCN to refine the embeddings of the nodes within the hyperedge. The Chebyshev spectral GCN leverages Chebyshev polynomial expansion and spectral graph theory to learn localized spectral filters. These filters enable the extraction of local and composite features from graphs that capture complex geometric structures. The model with code provided is here.

- HYPERSAGCN [26]: Self-attention-based graph convolutional network for hypergraphs (HyperSAGCN) utilizes a Spectral Aggregated Graph Convolutional Network (SAGCN) to refine the embeddings of nodes within each hyperedge. HyperSAGCN generates initial node embeddings by hypergraph random walks and combines node embeddings by MEAN (.) pooling to compute the embedding of hyperedge. The model with code provided is here.

- NHP [87]: Neural Hyperlink Predictor (NHP), is an enhanced version of HyperSAGCN. NHP initializes node embeddings using Node2Vec on the CE graph and then uses a novel maximum minimum-based pooling function that enables adaptive weight learning in a task-specific manner, incorporating additional prior knowledge about the nodes. The model with code provided is here.

- HPLSF [89]: Hyperlink Prediction using Latent Social Features (HPLSF) is a probabilistic method. It leverages the homophily property of the networks and introduces a latent feature learning approach, incorporating the use of entropy in computing hyperedge embedding. The model with code provided is here.

- HPRA [88]: Hyperlink Prediction Using Resource Allocation (HPRA) is a hyperedge prediction method based on the resource allocation process. HPRA calculates a hypergraph resource allocation (HRA) index between two nodes, taking into account direct connections and shared neighbors. The HRA index of a candidate hyperedge is determined by averaging all pairwise HRA indices between the nodes within the hyperedge. The model with code provided is here.

- CE-CAW: This model is a baseline that we apply CAW [33] on the CE of the hypergraph. CAW is a temporal edge prediction method that uses causal anonymous random walks to capture the dynamic laws of the network in an inductive manner. The model with code provided is here.

- CE-EVOLVEGCN: This is a snapshot-based temporal graph learning method that we apply EVOLVEGCN [90], which uses RNNs to estimate the GCN parameters for the future snapshots, on the CE of the hypergraph. The model with code provided is here.

- CE-GCN: We apply Graph Convolutional Networks [91] to the CE of the hypergraph to obtain node embeddings. Next, we use MLP to predict edges. The implementation is provided in the Pytorch Geometric library.

Table 4: Hyperparameters used in the grid search.

| Datasets | Sampling Number $M$ | Sampling Time Bias $\alpha$ | SetWalk Length $m$ | Hidden dimensions |
|---|---|---|---|---|
| NDC Class | 4, 8, 16, 32, 64, 128 | $\{0.5, 2.0, 20, 200\} \times 10^{-7}$ | 2, 3, 4, 5 | 32, 64, 128 |
| High School | 4, 8, 16, 32, 64, 128 | $\{0.5, 2.0, 20, 200\} \times 10^{-7}$ | 2, 3, 4, 5 | 32, 64, 128 |
| Primary School | 4, 8, 16, 32, 64, 128 | $\{0.5, 2.0, 20, 200\} \times 10^{-7}$ | 2, 3, 4, 5 | 32, 64, 128 |
| Congress Bill | 8, 16, 32, 64 | $\{0.5, 2.0, 20, 200\} \times 10^{-7}$ | 2, 3, 4, 5 | 32, 64, 128 |
| Email Enron | 8, 16, 32, 64 | $\{0.5, 2.0, 20, 200\} \times 10^{-7}$ | 2, 3, 4, 5 | 32, 64, 128 |
| Email Eu | 8, 16, 32, 64 | $\{0.5, 2.0, 20, 200\} \times 10^{-7}$ | 2, 3, 4 | 32, 64, 128 |
| Question Tags M | 8, 16, 32, 64 | $\{0.5, 2.0, 20, 200\} \times 10^{-7}$ | 2, 3, 4 | 32, 64, 128 |
| Users-Threads | 8, 16, 32, 64 | $\{0.5, 2.0, 20, 200\} \times 10^{-7}$ | 2, 3, 4 | 32, 64, 128 |
| NDC Substances | 8, 16, 32, 64 | $\{0.5, 2.0, 20, 200\} \times 10^{-7}$ | 2, 3, 4 | 32, 64, 128 |
| Question Tags U | 8, 16, 32, 64 | $\{0.5, 2.0, 20, 200\} \times 10^{-7}$ | 2, 3, 4 | 32, 64, 128 |

For node classification, we use additional five state-of-the-art deep hypergraph learning methods and a CE-based baseline:

- HyperGCN [19]: This is a generalization of GCNs to hypergraphs, where it uses hypergraph Laplacian to define convolution.

- AllDeepSets and AllSetTransformer [25]: These two methods are two variants of the general message passing framework, Allset, on hypergraphs, which are based on the aggregation of messages from nodes to hyperedges and from hyperedges to nodes.

- UniGCNII [28]: Is an advanced variant of UniGnn, a general framework for message passing on hypergraphs.

- ED-HNN [124]: Inspired by hypergraph diffusion algorithms, this method uses star expansions of hypergraphs with standard message passing neural networks.

- CE-GCN: We apply Graph Convolutional Networks [91] to the CE of the hypergraph to obtain node embeddings. Next, we use MLP to predict the labels of nodes. The implementation is provided in the Pytorch Geometric library. [91]

For all the baselines, we set all sensitive hyperparameters (e.g., learning rate, dropout rate, batch size, etc.) to the values given in the paper that describes the technique. Following [60], for deep learning methods, we tune their hidden dimensions via grid search to be consistent with what we did for CAt-Walk. We exclude HPLSF [89] and HPRA [88] from inductive hyperedge prediction as it does not apply to them.

### F.3 Implementation and Training Details

In addition to hyperparameters and modules (activation functions) mentioned in the main paper, here, we report the training hyperparameters of CAt-Walk: On all datasets, we use a batch size of 64 and set learning rate $= 10^{-4}$. We also use an early stopping strategy to stop training if the validation performance does not increase for more than 5 epochs. We use the maximum training epoch number of 30 and dropout layers with rate $= 0.1$. Other hyperparameters used in the implementation can be found in the README file in the supplement.

Also, for tuning the model's hyperparameters, we systematically tune them using grid search. The search domains of each hyperparameter are reported in Table 4. Note that, the last column in Table 4 reports the search domain for hidden dimensions of modules in CAt-Walk, including SetMixer, MLP-Mixer, and MLPs. Also, we tune the last layer pooling strategy with two options: SetMixer or Mean(.) whichever leads to a better performance.

We implemented our method in Python 3.7 with *PyTorch* and run the experiments on a Linux machine with *nvidia RTX A4000* GPU with 16GB of RAM.

Table 5: Performance on node classification: Mean ACC (%) ± standard deviation. Boldfaced letters shaded blue indicate the best result, while gray shaded boxes indicate results within one standard deviation of the best result.

| | Methods | High School | Primary School | Average Performance |
|---|---|---|---|---|
| Inductive | CE-GCN | 76.24 ± 2.99 | 79.03 ± 3.16 | 77.63 ± 3.07 |
| | HyperGCN | 83.91 ± 3.05 | 86.17 ± 3.40 | 85.04 ± 3.23 |
| | HyperSAGCN | 84.89 ± 3.80 | 82.13 ± 3.69 | 83.51 ± 3.75 |
| | AllDeepSets | 85.67 ± 4.17 | 81.43 ± 6.77 | 83.55 ± 5.47 |
| | UniGCNII | 88.36 ± 3.78 | 88.27 ± 3.52 | 88.31 ± 3.63 |
| | AllSetTransformer | **91.19 ± 2.85** | 90.00 ± 4.35 | 90.59 ± 3.60 |
| | ED-HNN | 89.23 ± 2.98 | 90.83 ± 3.02 | 90.03 ± 3.00 |
| | CAt-Walk | 88.99 ± 4.76 | **93.28 ± 2.41** | **91.13 ± 3.58** |
| Transductive | CE-GCN | 78.93 ± 3.11 | 77.46 ± 2.97 | 78.20 ± 3.04 |
| | HyperGCN | 84.90 ± 3.59 | 85.23 ± 3.06 | 85.07 ± 3.33 |
| | HyperSAGCN | 84.52 ± 3.18 | 83.27 ± 2.94 | 83.90 ± 3.06 |
| | AllDeepSets | 85.97 ± 4.05 | 80.20 ± 10.18 | 83.09 ± 7.12 |
| | UniGCNII | 89.16 ± 4.37 | 90.29 ± 4.01 | 89.73 ± 4.19 |
| | AllSetTransformer | 90.75 ± 3.13 | 89.80 ± 2.55 | 90.27 ± 2.84 |
| | ED-HNN | **91.41 ± 2.36** | 91.74 ± 2.62 | 91.56 ± 2.49 |
| | CAt-Walk | 90.66 ± 4.96 | **93.20 ± 2.45** | **91.93 ± 3.71** |

# G    Additional Experimental Results

## G.1    Results on More Datasets

Due to the space limit, we report the AUC results on only eight datasets in Section 4. Table 6 reports both AUC and average precision (AP) results on all 10 datasets in both inductive and transductive hyperedge prediction tasks.

## G.2    Node Classification

In the main text, we focus on the hyperedge prediction task. Here we describe how CAt-Walk can be used for node classification tasks.

For each node $u_0$ in the training set, we sample $\max\{\deg(u_0), 10\}$ hyperedges such as $e_0 = \{u_0, u_1, \ldots, u_k\}$. Next, for each sampled hyperedge we sample $M$ SetWalks with length $m$ starting from each $u_i \in e_0$ to construct $\mathcal{S}(u_i)$. Next, we anonymize each hyperedge that appears in at least one SetWalk in $\bigcup_{i=0}^{k} \mathcal{S}(u_i)$ by Equation 3 and then use the MLP-Mixer module to encode each Sw $\in \bigcup_{i=0}^{k} \mathcal{S}(u_i)$. To encode each node $u_i \in e_0$, we use Mean(.) pooling over SetWalks in $\mathcal{S}(u_i)$. Finally, for node classification task, we use a 2-layer perceptron over the node encodings to make the final prediction.

Table 5 reports the results of dynamic node classification tasks on High School and Primary School datasets. CAt-Walk achieves the best or on-par performance on dynamic node classification tasks. While all baselines are specifically designed for node classification tasks, CAt-Walk achieves superior results due to ① its ability to incorporate temporal properties (both from SetWalks and our time encoding module), which helps to learn underlying dynamic laws of the network, and ② its two-step set-based anonymization process that hides node identities from the model. Accordingly, CAt-Walk can learn underlying patterns needed for the node classification task, instead of using node identities, which might cause memorizing vertices.

## G.3    Performance in Average Precision

In addition to the AUC, we also compare our model with baselines with respect to Average Precision (AP). Table 6 reports both AUC and AP results on all 10 datasets in inductive and transductive hyperedge prediction tasks. As discussed in Section 4, CAt-Walk due to its ability to capture both temporal and higher-order properties of the hypergraphs, achieves superior performance and outperforms all baselines in both transductive and inductive settings with a significant margin.

Table 6: Performance on hyperedge prediction: AUC and Average Precision (%) ± standard deviation. Boldfaced letters shaded blue indicate the best result, while gray shaded boxes indicate results within one standard deviation of the best result. N/A: the method has computational issues.

| | Datasets | NDC Class | | High School | | Primary School | | Congress Bill | | Email Enron | |
|---|---|---|---|---|---|---|---|---|---|---|---|
| | Metric | AUC | AP | AUC | AP | AUC | AP | AUC | AP | AUC | AP |
| | | | | | | Strongly Inductive | | | | | |
| **Inductive** | CE-GCN | 52.31 ± 2.99 | 54.33 ± 2.48 | 60.54 ± 2.06 | 59.92 ± 2.25 | 52.34 ± 2.75 | 56.41 ± 2.06 | 49.18 ± 3.61 | 53.85 ± 3.92 | 63.04 ± 1.80 | 57.70 ± 2.27 |
| | CE-EvolveGCN | 49.78 ± 3.13 | 55.24 ± 3.56 | 46.12 ± 3.83 | 52.87 ± 3.48 | 58.01 ± 2.56 | 55.68 ± 2.41 | 54.00 ± 1.84 | 50.27 ± 1.76 | 57.31 ± 4.19 | 54.52 ± 3.79 |
| | CE-CAW | 76.45 ± 0.29 | 78.58 ± 1.32 | 83.73 ± 1.42 | 82.96 ± 1.04 | 80.31 ± 1.46 | 82.84 ± 1.71 | 75.38 ± 1.25 | 77.19 ± 1.38 | 70.81 ± 1.13 | 72.07 ± 1.52 |
| | NHP | 70.53 ± 4.95 | 68.18 ± 4.31 | 65.29 ± 3.80 | 62.86 ± 3.74 | 70.86 ± 3.42 | 71.31 ± 3.51 | 69.82 ± 2.19 | 64.09 ± 2.87 | 49.71 ± 6.09 | 50.01 ± 4.87 |
| | HyperSAGCN | 79.05 ± 2.48 | 77.24 ± 2.05 | 88.12 ± 3.01 | 82.72 ± 2.93 | 80.13 ± 1.38 | 76.32 ± 2.96 | 79.51 ± 1.27 | 80.58 ± 2.61 | 73.09 ± 2.60 | 72.29 ± 3.69 |
| | CHESHIRE | 72.24 ± 2.63 | 70.31 ± 2.26 | 82.54 ± 0.88 | 80.34 ± 1.19 | 77.26 ± 1.01 | 77.72 ± 0.76 | 79.43 ± 1.58 | 78.63 ± 1.25 | 70.03 ± 2.55 | 72.97 ± 1.81 |
| | CAt-Walk | **98.89 ± 1.82** | **98.97 ± 1.69** | **96.03 ± 1.50** | **96.41 ± 0.70** | **95.32 ± 0.89** | **96.03 ± 0.84** | **93.54 ± 0.56** | **93.93 ± 0.36** | **73.45 ± 2.92** | **74.66 ± 3.87** |
| | | | | | | Weakly Inductive | | | | | |
| | CE-GCN | 51.80 ± 3.29 | 50.94 ± 3.77 | 50.33 ± 3.40 | 48.54 ± 3.92 | 52.19 ± 2.54 | 53.21 ± 3.59 | 52.38 ± 2.75 | 50.81 ± 2.68 | 50.81 ± 2.87 | 55.38 ± 2.79 |
| | CE-EvolveGCN | 55.39 ± 5.16 | 57.24 ± 4.98 | 57.85 ± 3.51 | 63.26 ± 4.01 | 51.50 ± 4.07 | 52.59 ± 4.53 | 55.63 ± 3.41 | 5.19 ± 3.56 | 45.66 ± 2.10 | 50.93 ± 2.57 |
| | CE-CAW | 77.61 ± 1.05 | 80.03 ± 1.65 | 83.77 ± 1.41 | 83.41 ± 1.19 | 82.98 ± 1.06 | 80.84 ± 1.57 | 79.51 ± 0.94 | 80.39 ± 1.07 | 80.54 ± 1.02 | 77.41 ± 1.28 |
| | NHP | 75.17 ± 2.02 | 77.23 ± 3.11 | 67.25 ± 5.19 | 66.73 ± 4.94 | 71.92 ± 1.83 | 72.30 ± 1.89 | 69.58 ± 4.07 | 72.48 ± 4.83 | 60.38 ± 4.45 | 55.62 ± 4.67 |
| | HyperSAGCN | 79.45 ± 2.18 | 80.32 ± 2.23 | 88.53 ± 1.26 | 87.26 ± 1.49 | 85.08 ± 1.45 | 86.84 ± 1.60 | 80.12 ± 2.00 | 73.48 ± 2.77 | 78.86 ± 0.63 | 79.14 ± 1.51 |
| | CHESHIRE | 79.03 ± 1.24 | 78.98 ± 1.17 | 88.40 ± 1.06 | 86.53 ± 1.82 | 83.55 ± 1.27 | 79.42 ± 2.03 | 79.67 ± 0.83 | 80.03 ± 1.38 | 74.53 ± 0.91 | 75.88 ± 1.14 |
| | CAt-Walk | **99.16 ± 1.08** | **99.33 ± 0.89** | **94.68 ± 2.37** | **96.54 ± 0.58** | **96.53 ± 1.39** | **96.83 ± 1.16** | **98.38 ± 0.21** | **98.48 ± 0.18** | 64.11 ± 7.96 | 67.68 ± 6.93 |
| **Transductive** | HPRA | 70.83 ± 0.01 | 67.40 ± 0.00 | 94.91 ± 0.00 | 89.17 ± 0.00 | 89.86 ± 0.06 | 88.11 ± 0.02 | 79.48 ± 0.03 | 77.16 ± 0.03 | 78.62 ± 0.00 | 76.74 ± 0.00 |
| | HPLSF | 76.19 ± 0.82 | 77.62 ± 1.42 | 92.14 ± 0.29 | 92.79 ± 0.15 | 88.57 ± 1.09 | 87.69 ± 1.61 | 79.31 ± 0.52 | 75.88 ± 0.43 | 75.73 ± 0.05 | 75.32 ± 0.08 |
| | CE-GCN | 66.83 ± 3.74 | 65.83 ± 3.61 | 62.99 ± 3.02 | 59.76 ± 3.78 | 59.14 ± 3.87 | 55.59 ± 3.46 | 64.42 ± 3.11 | 63.19 ± 3.34 | 58.06 ± 3.80 | 55.27 ± 3.12 |
| | CE-EvolveGCN | 67.08 ± 3.51 | 66.51 ± 3.80 | 65.19 ± 2.26 | 59.27 ± 2.19 | 63.15 ± 1.32 | 65.18 ± 1.89 | 69.30 ± 2.27 | 64.38 ± 2.66 | 69.98 ± 5.38 | 67.76 ± 5.16 |
| | CE-CAW | 76.30 ± 0.84 | 77.73 ± 1.42 | 81.63 ± 0.97 | 79.37 ± 0.53 | 86.53 ± 084 | 87.03 ± 1.15 | 76.99 ± 1.02 | 77.05 ± 1.14 | 79.57 ± 0.14 | 78.37 ± 1.15 |
| | NHP | 82.39 ± 2.81 | 80.72 ± 2.04 | 76.85 ± 3.08 | 75.37 ± 3.12 | 80.04 ± 3.42 | 80.24 ± 3.49 | 80.27 ± 2.53 | 77.82 ± 1.91 | 63.17 ± 3.79 | 66.87 ± 3.19 |
| | HyperSAGCN | 80.76 ± 2.64 | 80.50 ± 2.73 | 94.98 ± 1.30 | 89.73 ± 1.21 | 90.77 ± 2.05 | 88.64 ± 2.09 | 82.84 ± 1.61 | 81.12 ± 1.79 | 83.59 ± 0.98 | 80.54 ± 1.66 |
| | CHESHIRE | 84.91 ± 1.05 | 82.24 ± 1.49 | 95.11 ± 0.94 | 94.29 ± 1.23 | 91.62 ± 1.18 | 92.72 ± 1.07 | 86.81 ± 1.24 | 83.66 ± 1.90 | 82.27 ± 0.86 | 81.39 ± 0.81 |
| | CAt-Walk | **98.72 ± 1.38** | **98.71 ± 1.36** | 95.30 ± 0.43 | 95.90 ± 0.44 | **97.91 ± 3.30** | **97.92 ± 2.95** | **88.15 ± 1.46** | **88.66 ± 1.57** | 80.47 ± 5.30 | **82.87 ± 3.50** |

| | Datasets | Email Eu | | Question Tags M | | Users-Threads | | NDC Substances | | Question Tags U | |
|---|---|---|---|---|---|---|---|---|---|---|---|
| | Metric | AUC | AP | AUC | AP | AUC | AP | AUC | AP | AUC | AP |
| | | | | | | Strongly Inductive | | | | | |
| **Inductive** | CE-GCN | 52.76 ± 2.41 | 50.37 ± 2.59 | 56.10 ± 1.88 | 54.15 ± 1.94 | 57.91 ± 1.56 | 59.45 ± 1.21 | 55.70 ± 2.91 | 54.29 ± 2.78 | 51.97 ± 2.91 | 55.03 ± 2.72 |
| | CE-EvolveGCN | 44.16 ± 1.27 | 49.15 ± 1.23 | 64.08 ± 2.75 | 60.64 ± 2.78 | 52.00 ± 2.32 | 52.69 ± 2.15 | 58.17 ± 2.24 | 57.35 ± 2.13 | 54.57 ± 2.25 | 57.16 ± 2.55 |
| | CE-CAW | 72.99 ± 0.20 | 73.45 ± 0.68 | 70.14 ± 1.89 | 70.26 ± 1.77 | 73.12 ± 1.06 | 72.64 ± 1.18 | 75.87 ± 0.77 | 73.19 ± 0.86 | 74.21 ± 2.04 | 76.52 ± 2.06 |
| | NHP | 65.35 ± 2.07 | 64.24 ± 1.61 | 68.23 ± 3.34 | 69.82 ± 3.41 | 71.83 ± 2.64 | 71.09 ± 2.83 | 70.43 ± 3.64 | 73.22 ± 3.03 | 72.52 ± 2.90 | 71.56 ± 2.26 |
| | HyperSAGCN | 78.01 ± 1.24 | 80.04 ± 1.87 | 73.66 ± 1.95 | 73.98 ± 1.35 | 73.94 ± 2.57 | 72.97 ± 2.45 | 75.85 ± 2.21 | 73.24 ± 2.75 | 78.88 ± 2.69 | 77.53 ± 2.28 |
| | CHESHIRE | 69.98 ± 2.71 | 70.10 ± 3.05 | N/A | N/A | 76.99 ± 2.92 | 74.03 ± 2.78 | 76.60 ± 2.19 | 74.91 ± 2.71 | 75.04 ± 3.39 | 75.46 ± 2.90 |
| | CAt-Walk | **91.68 ± 2.78** | **91.75 ± 2.82** | **88.03 ± 3.38** | **88.46 ± 3.09** | **89.84 ± 6.02** | **91.58 ± 4.37** | **93.29 ± 1.55** | **94.26 ± 1.21** | **97.59 ± 2.21** | **97.71 ± 2.07** |
| | | | | | | Weakly Inductive | | | | | |
| | CE-GCN | 49.60 ± 3.96 | 55.01 ± 3.25 | 55.13 ± 2.76 | 51.48 ± 2.66 | 57.06 ± 3.16 | 58.37 ± 2.86 | 60.92 ± 2.81 | 55.93 ± 2.03 | 56.85 ± 2.73 | 57.19 ± 2.52 |
| | CE-EvolveGCN | 52.44 ± 2.38 | 50.61 ± 2.32 | 61.79 ± 1.63 | 59.61 ± 1.12 | 55.81 ± 2.54 | 50.63 ± 2.46 | 58.48 ± 2.49 | 55.90 ± 2.51 | 54.10 ± 1.21 | 56.13 ± 2.32 |
| | CE-CAW | 73.54 ± 1.19 | 74.10 ± 1.41 | 77.29 ± 0.86 | 77.67 ± 1.94 | 80.79 ± 0.82 | 81.88 ± 0.63 | 77.28 ± 1.30 | 79.24 ± 1.19 | 76.51 ± 1.26 | 77.17 ± 1.39 |
| | NHP | 67.19 ± 4.33 | 66.53 ± 4.21 | 70.46 ± 3.52 | 65.66 ± 3.94 | 76.44 ± 1.90 | 75.23 ± 3.96 | 73.37 ± 3.51 | 70.62 ± 3.71 | 78.15 ± 4.41 | 79.64 ± 4.32 |
| | HyperSAGCN | 77.26 ± 2.09 | 74.05 ± 2.12 | 78.15 ± 1.41 | 76.19 ± 1.53 | 75.38 ± 1.43 | 70.35 ± 1.63 | 80.82 ± 2.18 | 76.67 ± 2.06 | 74.22 ± 1.91 | 70.57 ± 1.02 |
| | CHESHIRE | 77.31 ± 0.95 | 76.01 ± 0.98 | N/A | N/A | 81.27 ± 0.85 | 82.96 ± 1.41 | 80.68 ± 1.31 | 80.78 ± 1.13 | 77.60 ± 1.57 | 79.48 ± 1.79 |
| | CAt-Walk | **91.98 ± 2.41** | **92.22 ± 2.40** | **90.28 ± 2.81** | **90.56 ± 2.62** | **97.15 ± 1.81** | **97.55 ± 1.49** | **95.65 ± 1.82** | **96.18 ± 1.52** | **98.11 ± 1.31** | **98.25 ± 1.13** |
| **Transductive** | HPRA | 72.51 ± 0.00 | 71.08 ± 0.00 | 83.18 ± 0.00 | 80.12 ± 0.00 | 70.49 ± 0.02 | 72.83 ± 0.00 | 77.94 ± 0.01 | 75.78 ± 0.01 | 81.05 ± 0.00 | 81.71 ± 0.00 |
| | HPLSF | 75.27 ± 0.31 | 77.95 ± 0.14 | 83.45 ± 0.93 | 82.29 ± 1.06 | 74.38 ± 1.11 | 73.81 ± 1.45 | 82.12 ± 0.71 | 84.51 ± 0.62 | 80.89 ± 1.51 | 75.62 ± 1.38 |
| | CE-GCN | 64.19 ± 2.79 | 65.93 ± 2.52 | 55.18 ± 5.12 | 55.84 ± 4.53 | 62.78 ± 2.69 | 59.71 ± 2.25 | 63.08 ± 2.19 | 65.37 ± 2.48 | 66.79 ± 2.88 | 60.51 ± 2.26 |
| | CE-EvolveGCN | 64.36 ± 4.17 | 66.98 ± 3.72 | 72.56 ± 1.72 | 69.38 ± 1.51 | 68.55 ± 2.26 | 67.86 ± 2.61 | 70.09 ± 3.42 | 66.37 ± 3.17 | 71.31 ± 2.92 | 70.36 ± 2.72 |
| | CE-CAW | 78.19 ± 1.10 | 77.95 ± 0.98 | 81.73 ± 2.48 | 83.27 ± 2.34 | 80.86 ± 0.45 | 80.57 ± 1.08 | 84.72 ± 1.65 | 84.93 ± 1.26 | 80.37 ± 1.77 | 83.14 ± 0.97 |
| | NHP | 78.90 ± 4.39 | 76.95 ± 5.08 | 79.14 ± 3.36 | 78.79 ± 3.15 | 82.33 ± 1.02 | 81.44 ± 1.53 | 81.38 ± 1.42 | 82.17 ± 1.38 | 78.99 ± 4.16 | 80.06 ± 4.33 |
| | HyperSAGCN | 79.61 ± 2.35 | 75.99 ± 2.23 | 84.07 ± 2.50 | 84.22 ± 2.43 | 79.62 ± 2.04 | 79.38 ± 2.55 | 85.07 ± 2.46 | 85.32 ± 2.20 | 85.18 ± 2.64 | 80.99 ± 3.04 |
| | CHESHIRE | 86.38 ± 1.23 | 87.39 ± 1.07 | N/A | N/A | 82.75 ± 1.99 | 81.96 ± 1.75 | 86.30 ± 1.57 | 83.18 ± 1.92 | 87.83 ± 2.15 | 88.62 ± 1.76 |
| | CAt-Walk | **96.74 ± 1.28** | **97.08 ± 1.20** | **91.63 ± 1.41** | **92.28 ± 1.26** | **93.51 ± 1.27** | **94.98 ± 0.98** | **90.64 ± 0.44** | **91.96 ± 0.41** | **96.59 ± 4.39** | **97.06 ± 3.72** |

## G.4 More Results on Rnn v.s. MLP-Mixer in Walk Encoding

Most existing methods on (temporal) random walk encoding see a walk as a sequence of vertices and uses sequence encoders like Rnns or Transformers to encode each walk. The main drawback of these methods is that they fail to directly process temporal walks with irregular gaps between timestamps. That is, sequential encoders can be seen as discrete approximations of dynamic systems; however, this discretization often fails if we have irregularly observed data [125].nper This is the main motivation of recent studies to develop methods on continuous-time temporal networks [34, 126]. Most of these methods are too complicated and sometimes fail to generalize [127]. In CAt-Walk, we suggest a simple architecture to encode temporal walks by a time-encoding module along with a Mixer module (see Section 3.4 for the details). In this part, we evaluate the power of our Mixer module and compare its performance when we replace it with Rnns [82]. Figure 7 reports the results on all datasets. We observe that using MLP-Mixer with the time-encoding module in CAt-Walk can always outperform CAt-Walk when we replace MLP-Mixer with a Rnn, and mostly this improvement is more on datasets with high variance in their timestamps. We relate this superiority to the importance of using continuous-time encoding instead of sequential encoders.

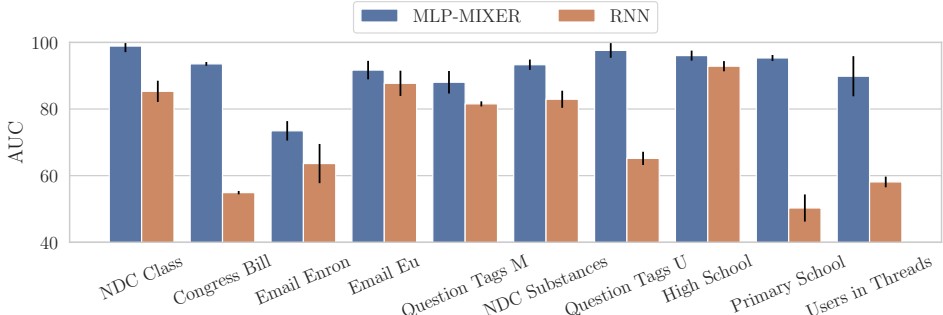

Figure 7: The importance of using MLP-Mixer in CAt-Walk. Using an Rnn instead of MLP-Mixer can damage the performance in *all* datasets. Rnns are sequential encoders and are not able to encode continuous time in the data.

# H Broader Impacts

Temporal hypergraph learning methods, such as CAt-Walk, benefit a wide array of real-world applications, including but not limited to social network analysis, recommender systems, brain network analysis, drug discovery, stock price prediction, and anomaly detection (e.g. bot detection in social media or abnormal human brain activity). However, there might be some potentially negative impacts, which we list as: ① Learning underlying biased patterns in the training data, which may result in stereotyped predictions. Since CAt-Walk learns underlying dynamic laws in the training data, given biased training data, the predictions of CAt-Walk can be biased. ② Also, powerful dynamic hypergraph models can be used for manipulation in the abovementioned applications (e.g., stock price manipulation). Accordingly, to prevent the potential risks in sensitive tasks, e.g., decision-making from graph-structured data in health care, interpretability and explainability of machine learning models on hypergraphs is a critical area for future work.

Furthermore, this work does not perform research on human subjects as part of the study and all used datasets are anonymized and publicly available.

