# OpenReview forum: "CAT-Walk: Inductive Hypergraph Learning via Set Walks"
_NeurIPS.cc/2023/Conference — NeurIPS 2023 poster_

### Official Review · Reviewer_eydA · 2023-07-06

**Soundness:** 3 good
**Presentation:** 4 excellent
**Contribution:** 3 good
**Rating:** 7
**Confidence:** 4

**Summary:**

The paper proposes an hyperedge learning method suitable for modelling temporal hyperedges. The approach consists of a novel hyperedge-centric random walk, a permutation-invariant pooling method and a technique for anonymizing both the identity of the nodes and  the identity of the hyperedges.

Different than standard random walks for hypergraphs that samples sequences of nodes, the SetWalk generates sequences of hyperedges, making it strictly more powerful than clique-expantion based methods. The pooling method used to encode the hyperedges represents a variation of MLP-mixer and the authors proved empirically that it is more powerful than other pooling approaches such as mean pooling and RNNs. Finally, the anonymization startegy incorporates positional information from hyperedges, while still allowing the model to be applied in the inductive setup. The experimental setup validates the benefits of applying the CAT-Walk for hyperedge prediction. Additionally, they also validate the importance of each component in a series of ablation studies.

**Strengths:**

* The proposed method contains a series of components that could be useful beyond the hyperedge prediction task presented in this paper. The idea of producing hyperedge-centric random walk is interesting and it is worth being explored in other contexts where random walk is required. Moreover the set-based pooling is a general technique that could be useful in other architectures where a set-level aggregator is needed.

* The paper is very nicely presented. Even if the proposed architecture contains a  complex pipeline, containing several components, each one with its own notations, the paper is easy to follow, with a nice structure and good visualisations.

* The paper introduces completely novel concepts and results presented in the experimental part clearly show the advantages of incorporating them in the method.

**Weaknesses:**

* SetWalk constitutes an hypergraph random walk technique  that, instead of producing sequences of nodes, it produces sequences of hyperedges. Given that, I am wondering what is the connection between the SetWalk method and a classical graph-based random walk applied on the line expansion of the hypergraph?

* The paper mentions that “[the proposed method] removes self-attention and RNNs from the walk encoding procedure, avoiding their limitations.” A detailed discussion about what are these limitations and how the proposed method overcomes them is required.

* The universal approximation results from [1] show that, by applying another function on top of the summation presented in Eq.4  $g(\sum_i f(x_i))$ leads to a universal approximator for multisets. How does SetMixer pooling compare to that in terms of expressivity? What are the advantages of using SetMixer instead of this classical approach?

* The paper mentions several times that a common drawbacks brought by various pooling methods is that they are losing important information. While being more expressive than some of them, it is not clear what are the limitations of SetMixer (if any). Is SetMixer universal approximator? If not, what are some examples of sets that SetMixer is not able to distinguish due to losing information.

* In Eq.5, shouldn’t the multiplication with W_s^2 be a right side multiplication. The way it is defined at this moment is not a permutation invariant function (which would contradict the results in Theorem 3).

* Theorem 4  states that the anonymization technique introduced in the paper is more expressive than any existing anonymization  strategies based on CE, mentioning that it can distinguish at least a pair of graphs that is not distinguished by those. However, being “more expressive than” should also imply that it does not lose any expressivity compared to CE.  However, the statement from Theorem 4 is unclear in this respect, since it is not obvious if the proposed method is able to distinguish between every pair of graphs that the previous methods are able to distinguish. This aspect should be more clearly mention in thetext.

* A couple of additional ablation studies should be included in order to clearly emphasize the advantages brought by CAT-Walk. A few suggestions include:

1. replacing the SetMIXER not only with RNNs but also self-attention/transformer as a more recent and powerful baseline.
2. Replacing the pooling with the sum-based universal approximator for sets  g(sum_i f(x_i)) where f and g are 2-layers MLPs seems a fairer comparison than the basic MEAN-pooling operator.

* In the ablation studies paragraph, which type of the hypergraph random walk is used for comparison? A citation or description of the method would be useful,

[1] How Powerful are Graph Neural Networks? Xu et al

**Questions:**

Please refer to the questions mentioned in the Weaknesses section.

**Limitations:**

A couple of limitations are mentioned in the paper.

---

> ### Author Rebuttal · Authors · 2023-08-08
>
> Thank you so much for your review! We really appreciate it! Please see below for our response to your comments:
>
> >*what is the connection between the SetWalk method and a classical graph-based random walk applied on the line expansion?*
> ----
> **Response:** Thank you for mentioning it. We have discussed this in Appendix C. There are two main challenges for random walks on hypergraphs:
> 1. Random walks are a sequence of pair-wise interconnected vertices, even though edges in a hypergraph connect sets of vertices.
> 2. A sampling probability of a walk on a hypergraph must be different from its sampling probability on the CE.
>
> Most existing works ignore (1) and focus on (2) to distinguish the walks on simple graphs and hypergraphs. Either (1) or (2) alone is not sufficient to take advantage of higher-order interactions. SetWalk aims to address both. Applying simple walks on line expansion graph ignores (2). As you mentioned, applying classical random walks on the line expansion can be seen as similar to SetWalks, but there are some important differences:
> - SetWalks are temporal walks that backtrack over time. Applying classical graph-based temporal walks on line expansion only captures the dynamics of nodes (edges in the line expansion graph), missing the temporal properties of the hyperedges in the main hypergraph.
> - Line expansion is equivalent to the CE of edge-to-vertex dual hypergraph. Applying a simple walk on the line expansion misses the higher-order dependencies of hyperedges in the hypergraph.
> - One might suggest applying a simple hypergraph walk on the edge-to-vertex dual hypergraph. However, it is always equivalent to clique expansion [1]. The sampling procedure of SetWalks (Eq. 1) guarantees hyperedge-dependent walk sampling on the dual hypergraph, which is more informative than simple (hypergraph) walks on the CE [1]. (We discuss this in Appendix D)
>
> We will add more explanations about it to make it clearer.
>
> ----
> > *... A detailed discussion about what are these limitations and how the proposed method overcomes them is required.*
> ----
> **Response:** Thank you for mentioning it. We have discussed this in Appendix G.5. The main drawback of these methods is that they fail to directly process temporal walks with irregular gaps between timestamps. Sequential encoders can be seen as discrete approximations of dynamic systems; however, this discretization often fails if we have irregularly observed data [2]. This motivates recent studies on continuous-time temporal graph neural networks. CAT-Walk by removing sequential encoders in the walk encoding process allows continuous-time temporal graph learning.
>
> We will add more explanations about the importance of removing sequential encoders to make this point clearer.
>
> ----
> > *How does SetMixer pooling compare to that in terms of expressivity? What are the advantages ...?*
> ----
> **Response:** To have a universal approximator for sets, we need
> 1. permutation-invariant property, and
> 2. showing the method can approximate any function.
>
> Theorem 3 shows the permutation-invariant property of SetMixer. Since SetMixer is an all-MLP architecture and MLPs are universal approximators, it is simple to show that SetMixer is also universal approximator.
> Thank you, we will add some explanations about the expressivity of SetMixer and mathematically prove it is a universal approximator.
>
> Also, one of the main advantages of SetMixer compared to classical approaches is its ability to capture cross-node dependencies of features. SetMixer uses the Softmax(.) function to bind token-wise information while a simple MLP or classical approach misses these token-wise dependencies. We also ran new ablation studies to show the importance of SetMixer (please see the attached file).
>
> ----
> > *... what are the limitations of SetMixer. Is SetMixer universal approximator? ...*
> ----
>
> **Response:** As we discussed above, SetMixer is a universal approximator. However, same as any other method, it might suffer from some limitations. Sometimes the importance of each element of the set is different. While SetMixer treats all the elements the same, future work can be to extend SetMixer to the case that the importance of all elements are not the same, possibly by an attention mechanism. We can add some explanations about these limitations.
>
> ----
> > *In Eq.5, shouldn’t the multiplication with W_s^2 be a right side multiplication.*
> ----
>
> **Response:** Thank you for mentioning it. Yes, it was a typo and it should be a right-side multiplication. Unfortunately, we found this typo after the main submission deadline and we could only fix it in the appendix. We will fix this typo in the final version.
>
> ----
> > *Theorem 4 states that the anonymization technique introduced ...*
> ----
>
> **Response:** Yes, sure, we will change the statement of Theorem 4 to make it clearer. In the proof, we also show our anonymization strategy is able to distinguish between every pair of graphs that the previous methods are able to distinguish.  The main idea for this is to decompose SetWalks into simple graph-based random walks and show that SetWalk includes the information provided by these walks. We will change the writing of Theorem 4 and its proof to make it clearer.
>
> ----
> > *A couple of additional ablation studies should be included ...*
> ----
> **Response:** Thank you for suggesting that. We have run the asked ablation studies and reported them in the attached file.
>
> ----
> >*... which type of hypergraph random walk is used?*
> ----
> **Response:** Yes, sure, we will add more explanations about the used random walk in the ablation study. In this experiment, we replace SetWalks with the most powerful existing hypergraph random walk, i.e., edge-dependent hypergraph random walk [1].
>
> Thank you so much once again for your time and review!
>
> [1] Random Walks on Hypergraphs with Edge-Dependent Vertex Weights. ICML 2019.
>
> [2] Neural controlled differential equations for irregular time series. NeurIPS 2020.

---

> > ### Comment · Reviewer_eydA · 2023-08-17
> > **Thanks for the response**
> >
> > Thanks for the detailed response. It effectively addressed my concerns. Therefore, I will increase my rating to Accept.

---

### Official Review · Reviewer_H3jP · 2023-07-06

**Soundness:** 3 good
**Presentation:** 3 good
**Contribution:** 3 good
**Rating:** 6
**Confidence:** 3

**Summary:**

The paper proposes an inductive approach to temporal hypergraph learning. The method includes two main components: a hypergraph random walk called SET-WALK and a pooling strategy termed SETMIXER, for extracting higher-order causal patterns. In addition, the authors design a simple neural network for hyperedge encoding.

**Strengths:**

- The paper is well-organized.
- The authors provide a new direction for the random walk on hypergraph, though the random walk seems more complicated than the CE-based random walk.
- The paper gives many intuitions and theoretical analyses to demonstrate the superior performance of set-walk compared to existing walks.
- The set-walk can be regarded as a generalization of node-walk on hypergraph, which seems interesting and useful.
- The experiments in Table 1 suggest the proposed methods significantly outperform the baselines.

**Weaknesses:**

1. The method contains several components: including Time Encoding, SetMixer, Setwalk, and MLP-Mixer. The experiments only provide three datasets in Table 2 to evaluate each component. It would be more convincing if each component could be evaluated in all datasets and all settings in Table 1.

2. Some definitions are not very easy to understand. It would be better to add some schematic for illustrating those concepts,  e.g. Def. 1 and Def. 2.

3. The experiments on node classification are inadequate. There are many important baselines and benchmarks concerning node classification, such as [1][2].

[1] Equivariant Hypergraph Diffusion Neural Operators

[2] Hypergraph Convolutional Networks via Equivalency Between Hypergraphs and Undirected Graphs

**Questions:**

Is any hypergraph random walk can replace the set-walk in the CAT-Walk network?

**Limitations:**

Yes.

---

> ### Author Rebuttal · Authors · 2023-08-08
>
> Thank you so much for your review! We really appreciate it! Please see below for our response to your comments:
>
> ----
> > *The method contains several components ... It would be more convincing if each component could be evaluated in all datasets and all settings in Table 1.*
>
> ----
>
> **Response:** Thank you for mentioning it. Running the ablation study on all datasets and in all settings requires #ablation-cases $\times$ #datasets $\times$ #settings $\times$ #run = $7 \times 10 \times 3 \times 10 = 2100$ experiments. Accordingly, in our ablation study, we followed existing studies (e.g., [1, 2, 3, 4]), and reported the results on a subset of datasets. However, we would be happy to evaluate each component of CAT-Walk on all datasets. We have started running asked experiments, and in the attached file we report the results on five datasets. The new results are compatible with previous findings and show the significance of each component in the CAT-Walk design. We can definitely and would be happy to add the results for other datasets to the final version of the paper. We also want to bring to your consideration that the results of the effect of MLP-Mixer (replacing MLP-Mixer with RNN) on all datasets have been reported in Figure 7 (page 35 in the appendix).
>
>
>
> ----
> > *Some definitions are not very easy to understand. It would be better to add some schematic for illustrating those concepts, e.g. Def. 1 and Def. 2.*
> ----
>
> **Response:** Thank you for your suggestion. Definition 1 defines temporal hypergraphs, and Definition 2 defines SetWalks. Figures 1, 2, and 3 include examples of temporal hypergraph (Def. 1) and SetWalks (Def. 2), but due to the space constraint, we didn’t directly illustrate these two definitions by referring to these figures. We will definitely add some examples to illustrate these definitions and make them clearer.
>
>
> ----
> > *The experiments on node classification are inadequate. There are many important baselines and benchmarks concerning node classification, such as [1][2].*
> ----
>
> **Response:** One of the significances of the CAT-Walk design is that it performs well in both hyperedge prediction and node classification tasks.  However,  one limitation to performing extensive experiments on node classification (as we have done for hyperedge prediction) is the lack of temporal hypergraph datasets with ground truth node labels. Both of the papers you mentioned have used static hypergraph datasets. However, in the attached file, we compare our CAT-Walk with the ED-HNN paper you mentioned (unfortunately, the code for the other paper you mentioned is not publicly available). We also want to bring to your consideration that in our node classification experiments, we have **seven** state-of-the-art node classification methods, even including very recent methods, which is a comparable number of baselines with the studies that only have focused on node classification tasks. Overall, in our experiments, we have **10 + 2** datasets and **8 + 7** ( **13** unique) baselines.
>
> We want to kindly bring to your consideration that coming up with one architecture suitable for different tasks is highly challenging and usually has not been done in the existing studies. Even in the existing studies in simple graphs, the state-of-the-art methods in each task differ significantly [5]. In hypergraph learning methods, for example, PhenomNN [6], AllSet, ED-HNN [7], and UniGNN are specifically designed for node classification tasks, and CHESHIRE, NHP, and HPRA are specifically designed for hyperedge prediction tasks. Accordingly, their experiments are also limited to a single task (either node classification or hyperedge prediction). CAT-Walk not only is suitable for both hyperedge prediction and node classification tasks, but it also outperforms baselines in both inductive and transductive settings.
>
>
> ----
> > *Is any hypergraph random walk can replace the set-walk in the CAT-Walk network?*
> ----
>
> **Response:** Yes, any hypergraph random walk can replace the SetWalk and can be used in the CAT-Walk framework. In our ablation study, we replace SetWalk with the state-of-the-art hypergraph random walk and show the significance of using SetWalk.
>
>
>
> Thank you so much once again for your time and review!
>
>
>
>
>
> ----
>
> [1] Hypergraph Convolutional Networks via Equivalency between Hypergraphs and Undirected Graphs. ICML 2022.
>
> [2] Inductive Representation Learning on Temporal Graphs. ICLR 2020.
>
> [3] NHP: Neural Hypergraph Link Prediction. CIKM 2020.
>
> [4] Inductive Representation Learning in Temporal Networks via Causal Anonymous Walks. ICLR 2021.
>
> [5] Open Graph Benchmark: Datasets for Machine Learning on Graphs. NeurIPS 2020.
>
> [6] From Hypergraph Energy Functions to Hypergraph Neural Networks. ICML 2023.
>
> [7] Equivariant Hypergraph Diffusion Neural Operators. ICLR 2023.

---

### Official Review · Reviewer_nNT7 · 2023-07-06

**Soundness:** 3 good
**Presentation:** 4 excellent
**Contribution:** 3 good
**Rating:** 7
**Confidence:** 4

**Summary:**

The paper proposes a walk-based learning method for temporal hyperedge prediction. Several conceptional contributions are made to walks in (temporal) hypergraphs which are compared to existing techniques such as clique or star expansion to represent hypergraphs by simple (weighted) graphs. A corresponding neural architecture is proposed and shown to outperform existing methods in an extensive experimental comparison.

**Strengths:**

* The newly introduced concepts and ideas on walks in hypergraphs are nicely motivated by the shortcomings of existing methods. The authors provide illustrative examples, which are discussed in sufficient detail.
* The paper is well-written and comprehensive. The appendix provides interesting additional results and answers most of my questions.
* The experimental comparison is extensive and shows good results for the proposed method.

**Weaknesses:**

* While the conceptional contributions are nicely motivated, this is not the case to the same extent for the design choices of the neural architecture. Some arguments remain vague or are not clear to me. Theorem 2 makes a statement on set functions (or pooling) and suggests that applying a function $f$ to the individual elements (or subsets of fixed size $k$) and computing their sum "can cause missing information." The statement is not precise, and the meaning of "$f$ is some function" is unclear. The proof in the appendix also is not helpful and essentially states that it is clear that the sum is less informative than the set itself. In light of the following papers [[Deep Sets](https://arxiv.org/abs/1703.06114), [Some Might Say All You Need Is Sum](https://arxiv.org/pdf/2302.11603.pdf)] this is not clear.
* The approach shows great results for the task of "temporal hyperedge prediction", which is very specific. For the standard task of node classification, it still works well but the improvement is less significant.

**Questions:**

* Can you make Theorem 2 more precise and address the above-mentioned concerns?
* For some datasets, it is not clear to me how hyperedges were derived. For example, face-to-face contacts in the high school dataset naturally give rise to simple graphs. Can you explain this?
* The considered task of "temporal hyperedge prediction" is very specific. Which competitors were designed for this task?

**Limitations:**

A detailed discussion of the prediction tasks the approach is (most) suitable for would strengthen the paper.

---

> ### Author Rebuttal · Authors · 2023-08-08
>
> Thank you so much for your review! We really appreciate it! Please see below for our response to your comments:
>
> ----
> >*While the conceptional contributions are nicely motivated, this is not the case to the same extent for the design choices ...*
> ----
>
> **Response:** Thank you for mentioning it. Yes, sure, we will make the statement in Theorem 2 more precise in the paper. The term “$f$ is some function” is used to say function $f$ is an arbitrary function that takes sets as input. Also, the term “missing information” refers to missing the higher-order information provided by the hypergraph representation. We say an approach (or a module) misses higher-order information if its output is the same when the input is either hypergraph $G$ or its projected graph (e.g., clique expansion). We will change the writing and mathematically formulate them to make both clearer.
>
> The main intuition of Theorem 2 is that a pooling function needs to capture higher-order
> dependencies of its input’s elements and if it can be decomposed into a summation of functions that capture only lower-order dependencies, then it misses higher-order information encoded in the data. In the proof of Theorem 2, we construct a hypergraph $G$ and show that the summarization of $G$’s projected graph, provided by this pooling function, is more informative than the provided summarization of $G$ itself by this pooling function. The term “sum is less informative than the set itself” means: Given a set $S = \\{a_1, …, a_k\\}$, we can construct any arbitrary algebraic combination of $a_1, …, a_k$, while $a_1 + a_2 + … + a_k$ is only one of them. Accordingly, given $q = a_1 + a_2 + … + a_k$, we do not have any information about the exact value of the higher-order combinations of $a_i$s, e.g., $p = a_1 a_2 + a_2 a_3 + … + a_{k - 1} a_k$. We can modify the writing of this proof to make this point clearer.
>
> Please note that results presented in Deep Sets correspond to the models that use “suitable transformations” on top of the summation. Our Theorem 2 only holds for pooling strategies that can be written in the form of Eq. 4 (e.g., SUM Pooling). Although using a simple linear layer (suitable transformations) on top of the summation might not cause missing higher-order information, it misses the cross-node dependencies of features. Our presented SetMixer uses the Softmax$(.)$ function to bind token-wise information in a non-parametric manner. We ran new ablation studies to show the importance of SetMixer and compared it with the sum-based universal approximator for sets (please see the attached file). These results show the significance of the architecture of SetMixer.
>
> ----
> > *The approach shows great results for the task of "temporal hyperedge prediction", which is very specific ...*
> ----
>
> **Response:** As we discussed in the paper, coming up with one architecture suitable for different tasks is highly challenging and usually has not been done in the existing studies. Even in the existing studies in simple graphs, the state-of-the-art methods in each task differ significantly [1]. In hypergraph learning methods, for example, PhenomNN [2], AllSet, ED-HNN [3], and UniGNN are specifically designed for node classification tasks, and CHESHIRE, NHP, and HPRA are specifically designed for hyperedge prediction tasks.
>
> One of the significances of the CAT-Walk is that it performs well in both hyperedge prediction tasks (with significant improvement) and node classification tasks (with improvement but less significant).
>
> ----
> > *Can you make Theorem 2 more precise and address the above-mentioned concerns?*
> ----
>
> **Response:** Yes, sure, we will make the statement in Theorem 2 more precise and clearer in the paper. We will change the writing and mathematically formulate the “missing information” term to make the theorem clearer. Please also see our response to your first comment.
>
> ----
> > *For some datasets, it is not clear to me how hyperedges were derived. For example, face-to-face contacts in the high school ... Can you explain this?*
> ----
>
> **Response:** The face-to-face contacts datasets (e.g., high school dataset) have recorded interactions at a resolution of 20 seconds. That is, a group of people are connected by a hyperedge if their minimum distance to each other during the past 20 seconds was less than a threshold. We also want to kindly bring to your consideration that all of the used datasets in this paper are previously released benchmark hypergraph datasets by other studies.
>
> We will add more explanations about each of the datasets and how they have been derived to make everything clearer.
>
> ----
> >  *Which competitors were designed for this task?*
> ----
>
> **Response:** All of the hypergraph learning competitors are designed for hyperedge prediction tasks. The only exception is HyperSAGCN, which can be used for both hyperedge prediction and node classification tasks. However, CAT-Walk significantly outperforms HyperSAGCN in both hyperedge prediction and node classification (please see the attached file) tasks.
>
> ----
> > *A detailed discussion of the prediction tasks ...*
> ----
>
> **Response:** Thank you for suggesting that. We believe that CAT-Walk is suitable for both node and hyperedge prediction tasks. Our datasets' domains vary from social, interaction, chemical, and communication networks. Due to the inductive nature of CAT-Walk, it can be applied to any network that changes based on specific dynamic laws and captures and learns these laws by sampling SetWalks. We would be happy to add a discussion in more detail about what type of domains are most suitable for CAT-Walk due to its design and architecture, as you mentioned.
>
> Thank you so much once again for your time and review!
>
> ----
> [1] Open Graph Benchmark: Datasets for Machine Learning on Graphs. NeurIPS 2020.
>
> [2] From Hypergraph Energy Functions to Hypergraph Neural Networks. ICML 2023.
>
> [3] Equivariant Hypergraph Diffusion Neural Operators. ICLR 2023.

---

> > ### Comment · Reviewer_nNT7 · 2023-08-16
> >
> > Thank you for answering my questions in detail.

---

### Official Review · Reviewer_BLYu · 2023-07-14

**Soundness:** 3 good
**Presentation:** 3 good
**Contribution:** 3 good
**Rating:** 7
**Confidence:** 3

**Summary:**

The paper introduces a novel way of defining random walks on hypergraph. The random walks consist in a sequence of hyperedges, where the transition probabilities take into account both the temporal closeness of the hyperedges and the importance of the nodes through which the hyperedges are connected.
The resulting random walk is more expressive (in the sense that transition probabilities capture more features of the hypergraph) than the Clique expansion and star expansion random graphs. These random walks are used to build an encoding of the nodes using a permutation invariant procedure. Finally these encodings are fed into a 2 layer MLP that outputs hyperedge predictions. The model compares favorably to other approaches used in the literature.

**Strengths:**

The paper is clearly written. Although I’m not completely familiar with the relevant literature, the method seems novel and aims to solve a relevant problem in hypergraph link prediction. The theoretical reasons for introducing a new type of random walk on hypergraphs are compelling, and the method indeed manages to solve the expressivity problems associated to earlier random walk approaches. The experimental results are well cured and show a significant improvement in performance over previous algorithms.

**Weaknesses:**

The theorems 1 and 2 are stated in an informal and imprecise way. While I understand the need not to flood the main text with mathematical details, one would have expected to find a rigorous formulation at least in the appendix. In partiuclar the definition of «expressiveness», «missing information» are not made explicit.

**Questions:**

Can you comment on the number of hyperparameters (such as $\alpha,M,m$) used in CAT-WALK compared to those needed in other algorithms? Is there more hyperparameter tuning to do when using CAT-WALK?


**Limitations:**

The authors have addressed the limitations of their algorithm.

---

> ### Author Rebuttal · Authors · 2023-08-08
>
> Thank you so much for your review! We really appreciate it! Please see below for our response to your comments:
>
> ----
> > *The theorems 1 and 2 are stated in an informal and imprecise way. While I understand the need not to flood the main text with mathematical details, one would have expected to find a rigorous formulation at least in the appendix. In partiuclar the definition of «expressiveness», «missing information» are not made explicit.*
> ----
>
> **Response:**  We followed the previous studies and use the commonly used graph isomorphism test as the measure of the expressive power of methods. That is, we say model $A$ is more expressive than model $B$, if
> 1.  for any pair of graphs like $(G_1, G_2)$ that $G_1 \neq G_2$, if model $B$ can distinguish $G_1$ and $G_2$ then model $A$ can also distinguish them,
> 2.  there is a pair of graphs like $(G’_1, G’_2)$ such that $A$ can distinguish them but $B$ cannot.
>
> With respect to “missing information,” we mean missing higher-order information provided by the hypergraph representation. That is, we say an approach (or a module) misses higher-order information if its output is the same when the input is either hypergraph $G$ or its projected graph (clique expansion).
>
> Thank you, yes, sure, we will explicitly and formally define both “expressiveness” and “missing information” in the paper as well as add additional explanations in the appendix.
>
>
>
>
> ----
> > *Can you comment on the number of hyperparameters (such as) used in CAT-WALK compared to those needed in other algorithms? Is there more hyperparameter tuning to do when using CAT-WALK?*
> ----
>
>
>
> **Response:** Thank you for asking about this! We will add more discussions about the number of hyperparameters used in CAT-Walk and other algorithms. In deep learning methods, in addition to the hidden dimensions, which are common hyperparameters in all algorithms, CAT-Walk uses **three** hyperparameters ($\alpha$, $m$, and $M$),  HYPERSAGCN uses **five** hyperparameters (window size, walk length, the number of walks per vertex, and $p$ and $q$ to control the tendencies that encourage outward exploration in walk sampling), and NHP uses **four** hyperparameters ($\lambda$ in the scoring function, number of GCN layers, and $p$ and $q$ to control the tendencies that encourage outward exploration in Node2Vec walk sampling). The only competitor algorithm with less number of hyperparameters is CHESHIRE with **two** hyperparameters ($K$: Chebyshev filter size, and dropout probability). Although CHESHIRE has less number of hyperparameters compared to CAT-Walk, the search space for the Chebyshev filter size is usually much larger than the search space for $m$ and $M$. Accordingly, depending on the task and dataset, either CAT-Walk or CHESHIRE might need less amount of hyperparameter tuning. We want to emphasize that CAT-Walk and CHESHIRE require a comparable amount of hyperparameter tuning while CAT-Walk achieves better AUC and Precision performance in the hyperedge prediction task.
>
>
>
> Thank you so much once again for your time and review!

---

> > ### Comment · Reviewer_BLYu · 2023-08-14
> >
> > Thank you for answering my questions! I found the answers very clear.

---

### Author Rebuttal · Authors · 2023-08-09

We want to thank all the reviewers for their time and comments. We are also grateful for all the constructive feedback for improving the paper.

We have attached a file with two tables. The first table reports the results of previous ablation studies and the two newly asked ablation studies (highlighted in blue) on five datasets. The second table reports the results of a new baseline for node classification tasks. The new results are compatible with the previous results, i.e.,

1. Ablation study results show the significance of each CAT-Walk component. While both the classical universal approximator and SetWalk are universal approximators, SetWalk architecture allows binding token-wise information, which captures the cross-feature dependencies of nodes. Also, replacing MLP-Mixer with Transformers damages the performance as sequential encoders consider time as a discrete variable while CAT-Walk uses a Time Encoding module along with MLP-Mixer to encode temporal walks, considering time as a continuous variable.

2. Not only CAT-Walk shows outstanding performance in hyperedge prediction tasks, but Table 2 also shows that CAT-Walk achieves competitive (the best or on-par) performance on node classification tasks.

---

### Decision · Program_Chairs · 2023-09-21

**Decision:**

Accept (poster)

**Comment:**

This paper was unanimously appreciated by the reviewers and is a nice contribution to the literature. The discussion was valuable and identified many points that were not clear from the manuscript. While these points were clarified in the discussion, the authors need to revise the paper to carry those clarifications into the manuscript.